# ATP and small amphiphilic molecules act as molecular matchmakers to fine-tune FET protein clusters
Mrityunjoy Kar ✉

FET (FUS-EWSR1-TAF15) family proteins form mesoscale clusters under physiological conditions at concentrations well below the threshold for phase separation. However, how ATP, an amphiphilic molecule and essential cellular metabolite, affects this clustering remains unclear. Here, I show that ATP modulates the size of subsaturation mesoscale clusters in a concentration-dependent manner. At low concentrations (1–2 mM), ATP promotes clustering by acting as a molecular crosslinker, leading to larger assemblies. At a moderate concentration (5 mM), clusters become smaller but remain stable, whereas at a higher concentration (10 mM), the cluster size is reduced. Other amphiphilic molecules, including sodium xylene sulfonate, sodium toluene sulfonate, and hexanediol, display comparable concentration-dependent effects. These observations cannot be explained solely by hydrotropic or kosmotropic mechanisms; instead, they arise from non-specific interactions between amphiphilic molecules and protein. Thus, the intrinsic chemical features of small molecules and FET proteins collectively govern mesoscale clustering at subsaturation concentrations.

FET proteins are classified as RNA-binding proteins, including Fused in Sarcoma (FUS), Ewing Sarcoma RNA-binding protein 1 (EWSR1), and TATA-Box Binding Protein Associated Factor 15 (TAF15)[1]. The FET proteins, present in every cell in our body, exhibit nucleocytoplasmic shuttling and play pivotal roles in RNA biogenesis, such as transcription, splicing, stability, transport, and metabolism[1]. Members of this protein family demonstrate phase separation phenomena in vivo and in vitro[2,3], influencing various biological processes, from DNA damage responses to cytoplasmic stress reactions[4–6]. Dysregulated phase transitions of FET proteins have been associated with cancers and neurodegenerative disorders[7]. FET proteins are present in cytoplasmic inclusions characteristic of Frontotemporal lobar degeneration (FTLD) and Amyotrophic lateral sclerosis (ALS)[8].

FET proteins spontaneously organize into diverse mesoscale molecular assemblies known as clusters, even at concentrations well below the saturation threshold for phase separation[9,10]. Similar behavior of sub-saturation cluster formation has been observed in other phase-separating proteins[11–14]. Recent investigations have unveiled the prevalence of biomacromolecular mesoscale structures within the cytoplasm[15]. Specifically, Li and colleagues have highlighted the functional significance of gas vesicle protein clusters in sub-saturated solutions, which impact bacterial fitness[13]. Given the relevance of sub-saturation mesoscale clusters in several phase-separating proteins, comprehending the impact of the most abundant cellular metabolite, adenosine triphosphate (ATP), on the distribution of sub-saturation mesoscale clusters of FET proteins is imperative.

ATP functions as the energy currency of biology[16,17], maintaining an average cellular concentration of 4.4 mM[18]. It releases energy through hydrolysis, converting into ADP and inorganic phosphate when needed[16]. Beyond energy provision, ATP is thought to regulate protein states via its hydrotrope property[19–26]. Hydrotropes are small, amphiphilic molecules that enhance the solubility of hydrophobic compounds in water by organizing around them through non-specific interactions, without forming micelles like surfactants[27,28]. ATP, regarded as a 'biological hydrotrope,' helps keep high concentrations of proteins soluble, prevents aggregation, and dissolves existing aggregates[19]. Additionally, cellular ATP levels influence proteostasis and impact the physicochemical properties of the cytoplasm, including viscosity, macromolecular crowding, and liquid-liquid phase separation[29–31]. This study investigates how varying ATP concentrations affect sub-saturated FET protein clusters, aiming to understand ATP's role in maintaining protein homeostasis within the cell.

In addition to ATP, this study explores how small amphiphilic molecules, sodium xylene sulfonate (NaXS), sodium toluene sulfonate (NaTS), and 1,6-hexanediol (HD) (Fig. 1) influence the formation of sub-saturation clusters of FET proteins. We investigate whether common hydrotropes and HD affect cluster formation similarly to ATP, which we previously found to prevent phase separation at saturation concentrations and maintain clusters[9]. Using techniques like dynamic light scattering (DLS) and

Leibniz-Institut für Polymerforschung Dresden e.V, Dresden, Germany. ✉e-mail: kar@ipfdd.de

**Fig. 1 | Amphiphilic small molecules.** Chemical structure drawing of various molecules, including adenosine triphosphate (ATP), sodium xylene sulfonate (NaXS), sodium toluene sulfonate (NaTS), and hexanediol (HD). In the drawing, the red represents the hydrophobic segments, while the blue represents the hydrophilic segments. Additionally, navy blue depicts more hydrophilic ionic moieties than hydroxy groups, shown in light blue. The NaXS is a mixture of two isomers, where two methyl groups are arranged in either ortho, meta, or para position in the benzene ring.

nanoparticle tracking analysis (NTA), we analyze mesoscale cluster formation with and without these molecules. Our data aim to reveal the general principles governing mesoscale clustering, considering the inherent chemistry of proteins and small molecules. Additionally, Nano differential scanning fluorimetry (NanoDSF) is employed to assess the temperature-dependent stability of FET proteins in the presence of these molecules. Overall, this study enhances understanding of the molecular interactions that regulate protein clustering and stability.

## Results

### ATP modulates the size distribution of sub-saturation FUS-SNAP clusters

To investigate how ATP and other small amphiphilic molecules influence the formation of mesoscale FET protein clusters at sub-saturation concentrations, we used FUS fused with a C-terminal SNAP-tag (FUS-SNAP). The SNAP-tag increases the saturation concentration of FUS compared to its untagged form, allowing for a broader concentration range to be studied[3,9]. We conducted DLS experiments to monitor sub-saturation cluster formation by measuring the hydrodynamic diameter over 32 min, following previous studies[9,10]. Figure 2a–c shows the hydrodynamic radius over time for FUS-SNAP at 0.25 μM, 0.5 μM, and 1 μM, respectively, in the presence of 10 mM KCl and 20 mM HEPES buffer (pH 7.4). To specifically examine how small amphiphilic molecules affect cluster formation, we maintained a low KCl concentration (10 mM), reflecting intracellular conditions[32]. We previously observed that higher KCl levels inhibit FET protein clustering, while metabolites such as potassium glutamate strongly

promote it[10]. In this study, potassium glutamate was intentionally omitted to isolate the effects of the amphiphilic molecules on clustering.

Our DLS data revealed a time-dependent increase in the mean hydrodynamic diameter of FUS-SNAP mesoscale clusters across all concentrations, with higher FUS-SNAP concentrations leading to a faster rate of cluster growth (Fig. 2a–c). Autocorrelation data (Supplementary Fig. 1) confirmed cluster size increases over time without slow modes indicative of cluster-cluster coalescence[9] at low salt concentrations. However, at 1 μM FUS-SNAP, the cluster size exceeded one micron over time, aligning with phase separation under these conditions, unlike the sub-saturation clusters observed at 0.25 μM and 0.5 μM. The average hydrodynamic diameters of FUS-SNAP clusters at 30 minutes were approximately 281 nm, 441 nm, and 1021 nm for 0.25 μM, 0.5 μM, and 1 μM FUS-SNAP, respectively (Fig. 2d).

To complement the DLS findings, we performed NTA, which provides both size distributions and particle concentrations. NTA results were comparable with DLS data, showing smaller cluster sizes at 0.25 μM and 0.5 μM FUS-SNAP compared to 1 μM (Fig. 2e, Supplementary Table 1). Notably, DLS and NTA data do not overlap for the same sample; NTA data revealed smaller scatterer populations than DLS data[33]. Volume fraction measurements at 0.2, 0.4, and 2.4 for 0.25, 0.5, and 1 μM FUS-SNAP, respectively (Fig. 2f), further indicated cluster formation at lower concentrations and phase separation at 1 μM in 10 mM KCl and 20 mM HEPES (pH 7.4).

For ATP-dependent studies, we used ATP·Mg²⁺, the biologically active complex formed by ATP binding to magnesium ions. At 1 mM and 2 mM ATP·Mg²⁺, FUS-SNAP clusters continued to grow over 32 minutes (Fig. 2a–c). Cluster size increased by 1.2- to 2.3-fold in 1 mM ATP·Mg²⁺ over 30 min compared to buffer-only conditions (Fig. 2d), depending on FUS-SNAP concentration. Notably, slow modes, suggestive of cluster coalescence[9], were not detected in the correlation functions (Supplementary Figs. 2–4), indicating that low ATP·Mg²⁺ concentrations promoted larger assemblies without cluster-cluster coalescence. When higher concentrations of ATP·Mg²⁺ were present, the cluster size was reduced compared to 1 and 2 mM ATP·Mg²⁺ at all FUS-SNAP concentrations (Fig. 2a–d). Notably, at 5 mM and 10 mM ATP·Mg²⁺, cluster growth plateaued around 30 min, indicating stabilization of clusters (Fig. 2a–c). At a sub-saturation concentration of 0.25 μM FUS-SNAP, NTA data mirrored the trends observed in DLS (Fig. 2g). A similar trend was observed in volume fraction data (Fig. 2h), underscoring the role of ATP in modulating mesoscale cluster formation.

Furthermore, we examined how different pH levels influence the modulation of FUS-SNAP clusters by ATP·Mg²⁺. At pH 5.5 and 7.4, the FUS-SNAP clusters are smaller compared to pH 8.2 under buffer-only conditions, similar to the earlier reported observation[9]. At pH 5.5, the cluster size remains largely unchanged with increasing concentrations of ATP·Mg²⁺ (Fig. 2i, Supplementary Fig. 5b). Conversely, at pH 8.2, the patterns are similar to those seen at pH 7.4 (Fig. 2i, Supplementary Fig. 5b), showing clear modulation of cluster size with higher concentrations of ATP·Mg²⁺.

As a control, we examined ATP without Mg²⁺ using 0.25 μM FUS-SNAP (Supplementary Fig. 6). The trends remained similar, although the effect required lower ATP concentrations than ATP·Mg²⁺. At 0.5 mM ATP, cluster size increased, whereas at 1 mM ATP, it decreased relative to buffer-only conditions (Supplementary Fig. 6a). DLS data were inconclusive at 5 and 10 mM ATP because clustering was significantly inhibited. Additionally, NTA data followed the expected trend: cluster size increased at 0.5 mM ATP and decreased at higher concentrations (Supplementary Fig. 6b). At lower ATP concentrations, volume fraction data aligned well with cluster size, showing an increase at 0.5 mM ATP and a decrease at 1 mM ATP compared to buffer-only conditions (Supplementary Fig. 6c). At 5 and 10 mM ATP, long tails with larger populations led to an overall increase in the volume fraction (Supplementary Fig. 6c).

### Regulation of sub-saturation FUS-SNAP clusters by ADP and AMP but not adenosine

ATP consists of three phosphate groups linked to the 5' carbon of the sugar molecule of adenosine (Fig. 1). To pinpoint the key moieties responsible for

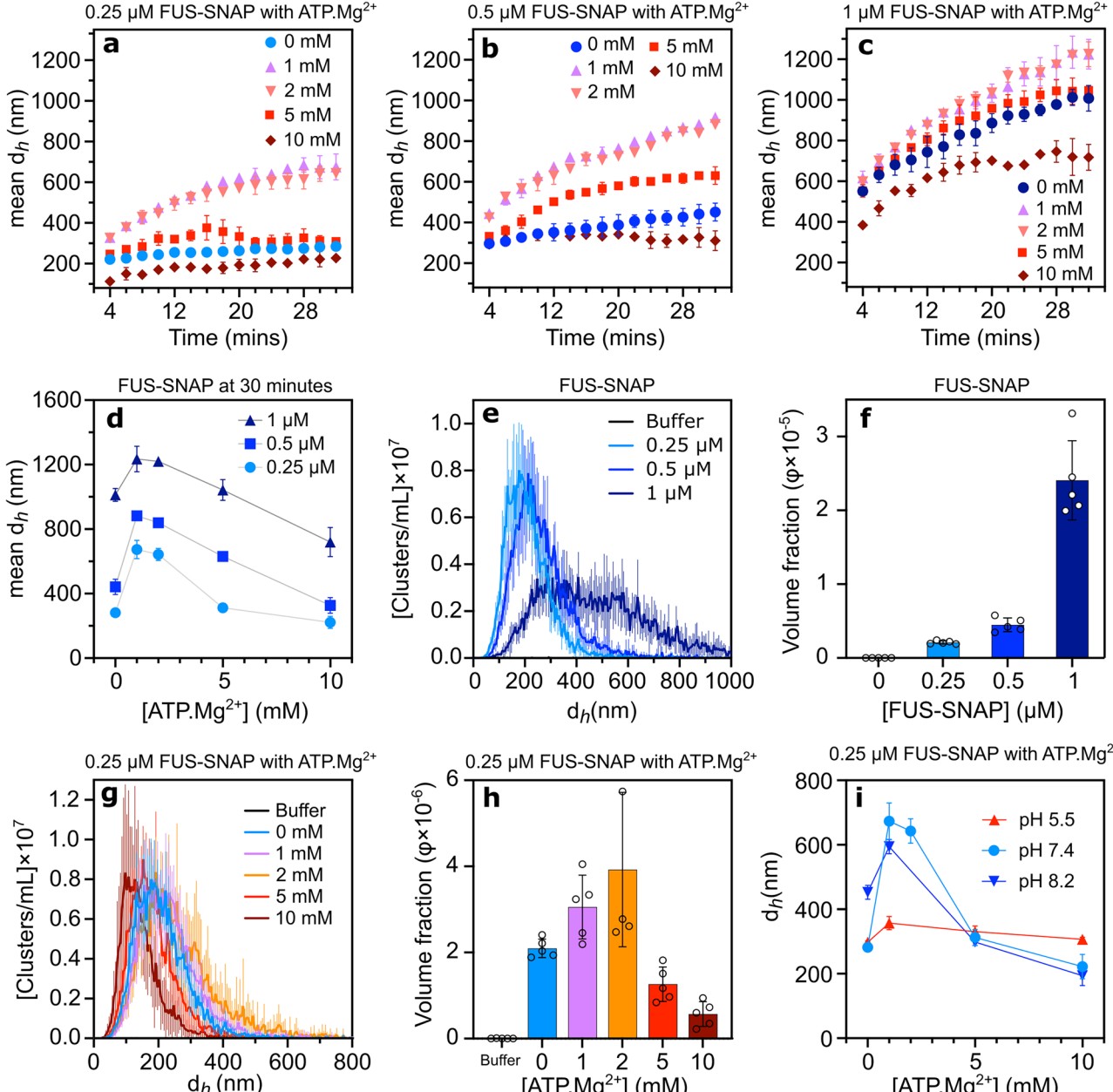

**Fig. 2 | ATP.Mg$^{2+}$ modulates the sub-saturation cluster size of FUS-SNAP.** Dynamic Light Scattering (DLS) data depict the mean hydrodynamic diameter ($d_h$) of mesoscale clusters at different sub-saturation concentrations of FUS-SNAP (0.25 μM (**a**), 0.5 μM (**b**), and 1 μM (**c**)) over 32 min with varying concentrations of ATP.Mg$^{2+}$. The mean hydrodynamic diameter ($d_h$) of 0.25 μM, 0.5 μM, and 1 μM FUS-SNAP at 30 min with varying ATP.Mg$^{2+}$ concentrations (**d**). Nanoparticle Tracking Analysis (NTA) shows $d_h$ of mesoscale clusters at 0.25 μM, 0.5 μM, and 1 μM FUS-SNAP (**e**) in 10 mM KCl and 20 mM HEPES pH 7.4. The volume fraction of mesoscale clusters is shown for different FUS-SNAP concentrations (**f**). NTA data for cluster' $d_h$ at 0.25 μM FUS-SNAP with different ATP.Mg$^{2+}$ concentrations are shown in (**g**), while (**h**) displays the corresponding volume fraction of clusters. The $d_h$ of 0.25 μM FUS-SNAP at 30 min with varying ATP.Mg$^{2+}$ concentrations at pH 5.5, 7.4, and 8.2 (**i**). NTA data are recorded at around 6–8 min from the time of sample preparation. Data are presented as mean ± SD, with $n$ = 3 (DLS) and $n$ = 5 (NTA) independent samples.

cluster regulation at 0.25 μM FUS-SNAP, we progressively removed phosphate groups from ATP and tested adenosine diphosphate (ADP), adenosine monophosphate (AMP), and adenosine (Fig. 3a). We observed trends similar to ATP.Mg$^{2+}$ in the presence of ADP at equivalent concentrations (Fig. 3b, e). At 1 mM AMP, cluster sizes increased compared to the buffer-only condition. However, AMP significantly inhibited cluster formation at higher concentrations (5 and 10 mM), resulting in obscure (low signal-to-noise ratio) DLS data (Fig. 3c, e). In contrast, adenosine-induced slight size increases at all concentrations compared to the buffer-only condition, with no discernible concentration-dependent trend (Fig. 3d, e).

Further investigation at 1 μM FUS-SNAP revealed similar trends (Supplementary Fig. 7a, b). In the case of adenosine, 1 μM FUS-SNAP cluster size increased slightly at equivalent concentrations of ATP.Mg$^{2+}$, no consistent size trends were observed (Supplementary Fig. 7c). As shown in Fig. 3f, cluster sizes increased at low concentrations but decreased with increasing concentrations of ADP and AMP (5 and 10 mM) compared to the 1 mM condition. Similar to the 0.25 μM, at 1 μM FUS-SNAP, the presence of adenosine also shows similar trends: the condensate size increases at all concentrations compared to the buffer-only condition, without a clear concentration-dependent pattern (Fig. 3f).

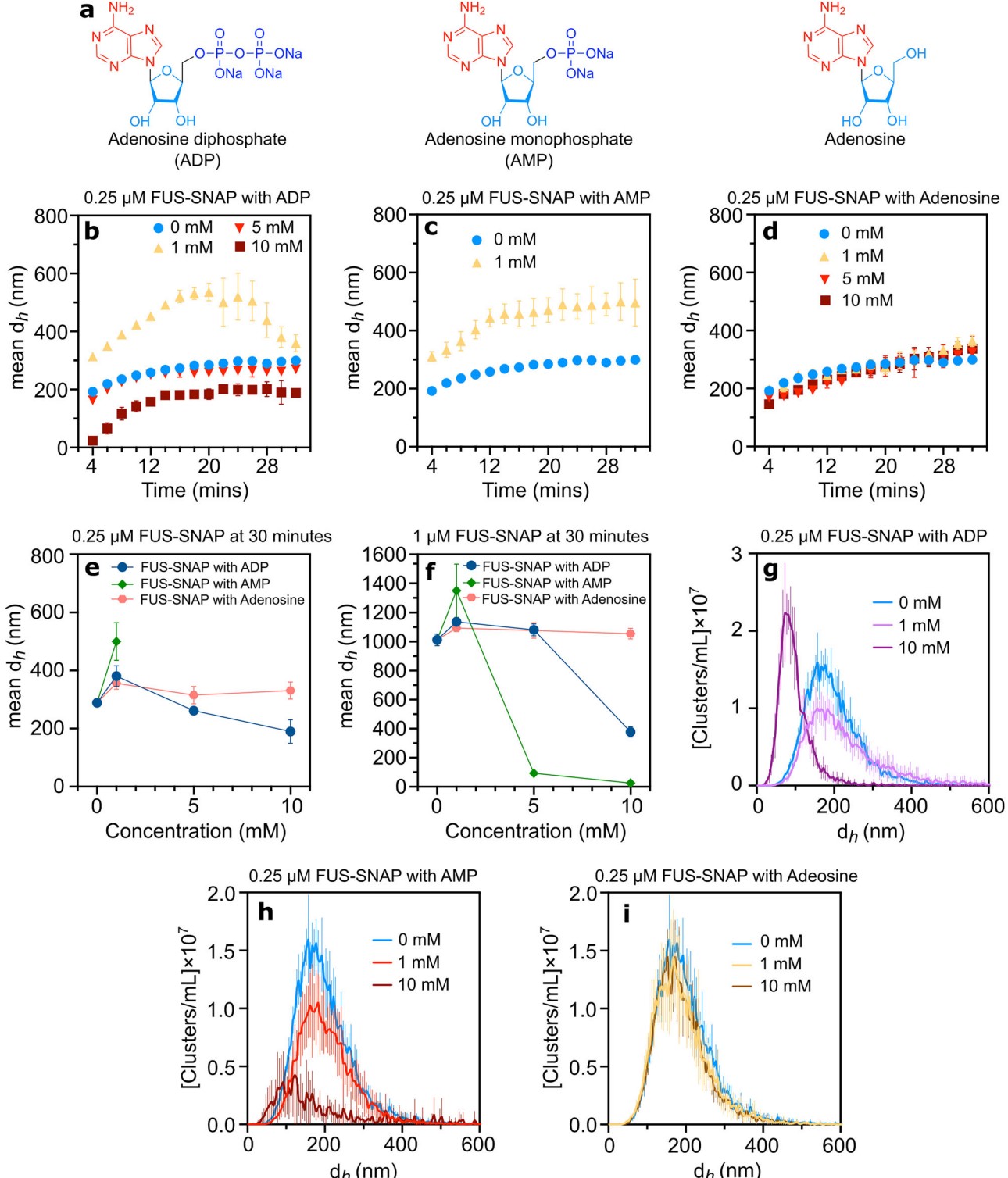

**Fig. 3 | Adenosine diphosphate (ADP) and adenosine monophosphate (AMP) modulate the size of sub-saturation FUS-SNAP clusters, whereas adenosine shows no regulatory effect.** Chemical structures of ADP, AMP, and adenosine (**a**). Dynamic Light Scattering (DLS) data showing mean hydrodynamic diameters ($d_h$) of mesoscale clusters at 0.25 µM FUS-SNAP over 32 minutes in the presence of various concentrations of ADP (**b**), AMP (**c**), and adenosine (**d**). DLS data shows a mean $d_h$ of 0.25 µM FUS-SNAP (**e**) and 1 µM FUS-SNAP (**f**) at 30-min time points with different concentrations of ADP, AMP, and adenosine. Nanoparticle Tracking Analysis (NTA) data at 0.25 µM FUS-SNAP for clusters' $d_h$ with ADP (**g**), AMP (**h**), and adenosine (**i**). NTA data are recorded at around 6–8 min from the time of sample preparation. Data are presented as mean ± SD from $n = 3$ (DLS) and $n = 5$ (NTA) independent experiments.

NTA data corroborated the DLS findings at 0.25 µM FUS-SNAP, revealing consistent trends. At 1 mM ADP, cluster size distributions showed a slight increase, whereas at 10 mM ADP, the distributions decreased compared to buffer-only conditions (Fig. 3g). The volume fraction followed a similar pattern, increasing at 1 mM ADP and decreasing at 10 mM ADP (Supplementary Fig. 8a). At 1 mM AMP, the cluster size distribution was slightly larger (Fig. 3h), though the volume fraction of clusters was lower than in the buffer-only condition (Supplementary Fig. 8b). At 10 mM AMP,

the cluster size distribution and volume fraction significantly decreased compared to buffer-only conditions (Supplementary Fig. 8b). In the presence of adenosine, no clear trends were observed in cluster size distributions at different concentrations (Fig. 3i). Additionally, the volume fractions at 1 mM and 10 mM adenosine remained similar (Supplementary Fig. 8c) but were reduced compared to buffer-only conditions. These observations underscore the essential role of phosphate moieties in regulating sub-saturation FUS-SNAP cluster formation.

### The phosphate group in nucleotides is crucial for controlling the size of sub-saturated FUS-SNAP clusters

To investigate how the phosphate group in ATP, ADP, and AMP influences cluster size, we examined the effects of sodium tripolyphosphate (STPP), sodium pyrophosphate (SPP), and sodium phosphate (SP) (Fig. 4a) on the sub-saturation cluster formation of FUS-SNAP. Notably, at low STPP, SPP, and SP concentrations, the hydrodynamic diameter of the 0.25 µM FUS-SNAP sub-saturation clusters increased over time, mirroring the effects of ATP.$Mg^{2+}$, ADP, and AMP when compared to buffer-only conditions (Fig. 4b–e). However, at higher concentrations (5 and 10 mM), the cluster size of 0.25 µM FUS-SNAP decreased and stabilized over time, in contrast to the buffer-only condition (Fig. 4b–e). The formation of clusters at 1 µM FUS-SNAP in the presence of STPP, SPP, and SP displayed similar trends to those observed at sub-saturation concentrations (Supplementary Fig. 9, Fig. f).

Furthermore, Fig. 4g illustrates the NTA data, showing the size distributions of 0.25 µM FUS-SNAP in the presence of 1 mM and 10 mM STPP. At 1 mM STPP, the size distribution shifted toward larger clusters, whereas at 10 mM STPP, the distribution narrowed, and the volume fraction of clusters decreased compared to the buffer-only condition (Supplementary Fig. 10a). A similar pattern was observed with 0.25 µM FUS-SNAP in the presence of SPP (Fig. 4h, Supplementary Fig. 10b). In contrast, with SP, the cluster size distributions remained relatively consistent across all concentrations, though increasing SP concentrations led to a decrease in the volume fraction of clusters (Fig. 4i, Supplementary Fig. 10c). This data suggests that the phosphate moieties (STPP, SPP, and SP) regulate FUS-SNAP cluster formation in a concentration-dependent manner.

### Similar to ATP.$Mg^{2+}$, common hydrotropes also regulate the size of sub-saturated FUS-SNAP clusters

To investigate further, we examined the effects of the common hydrotropes, NaXS and NaTS, on sub-saturation cluster formation. As illustrated in Fig. 5a, the hydrodynamic diameters of 0.25 µM and 1 µM FUS-SNAP clusters in the presence of hydrotropes exhibit trends similar to ATP.$Mg^{2+}$. At lower concentrations, these hydrotropes promote an increase in cluster size, while at higher concentrations, the cluster size decreases compared to buffer-only conditions (Supplementary Fig. 11). However, the concentration ranges required for NaXS and NaTS to produce these effects are significantly higher than those required for ATP.$Mg^{2+}$. At lower FUS-SNAP concentrations (0.25 µM), cluster inhibition was observed at 40 mM NaXS and NaTS (Fig. 5a, Supplementary Fig. 11a, b). For 1 µM FUS-SNAP, inhibition occurred at around 60 mM (Fig. 5a, Supplementary Fig. 11c, d).

Additionally, NTA data correlated well with DLS measurements at a concentration of 0.25 µM FUS-SNAP. At 10 mM NaXS, the cluster size distribution shifted toward larger sizes, while at 40 mM NaXS, the cluster size decreased significantly compared to the buffer-only condition (Fig. 5b). The cluster volume fraction followed a similar pattern: a higher value at 10 mM NaXS and a marked decrease at 40 mM NaXS (Supplementary Fig. 12a). NaTS exhibited comparable behavior (Fig. 5c, Supplementary Fig. 12b) to NaXS. These findings demonstrate that common hydrotropes, like NaXS and NaTS, mimic the cluster-regulating effects of ATP.$Mg^{2+}$ but require higher concentration ranges to exert similar effects.

### Hexanediol regulates the size of FUS-SNAP clusters

We next investigated HD due to its amphiphilic structural similarity to ATP and hydrotropes (Fig. 1). HD consists of six linked methylene groups

(hydrophobic) with hydroxyl moieties (hydrophilic) at both ends (Fig. 5d). To compare its effects, we also tested triethylene glycol (TEG), which has the same number of methylene groups as HD but with oxygen atoms separating each ethylene unit, thereby reducing hydrophobicity (Fig. 5d).

Interestingly, HD and TEG exhibited different trends of effects depending on FUS-SNAP concentrations. At sub-saturation concentration (0.25 µM FUS-SNAP), HD and TEG slightly increased cluster size with increasing concentrations compared to the buffer-only condition (Supplementary Fig. 13a, b, Fig. 5e). Conversely, at the phase separation concentration (1 µM FUS-SNAP), higher concentrations of HD and TEG led to a significant decrease in cluster size compared to the buffer-only condition (Fig. 5e). Notably, HD had a more substantial effect on cluster size reduction than TEG. At 0.5 and 1 M concentrations for both compounds, the cluster size diminished and stabilized compared to lower concentrations and the buffer-only condition (Supplementary Fig. 13c, d).

NTA data corroborated the DLS findings. As shown in Fig. 5f, g, increasing concentrations of HD and TEG, respectively, caused a shift toward larger cluster size distributions at 0.25 µM FUS-SNAP compared to buffer-only conditions. The volume fraction of clusters followed a similar pattern (Supplementary Fig. 14a, b). These results indicate that significantly high concentrations of HD and TEG regulate FUS-SNAP clusters at higher protein concentrations. However, HD and TEG show limited or no effect on lower concentrations of FUS-SNAP clusters.

### The formation of sub-saturated clusters of untagged FUS with ATP and hydrotropes shows similar patterns to FUS-SNAP

To confirm that the SNAP-tag does not alter the clustering behavior of untagged FUS in the presence of small molecules, we examined sub-saturation cluster formation using untagged FUS at varying concentrations of ATP.$Mg^{2+}$ and other small molecules. In the presence of 1 mM ATP.$Mg^{2+}$, the hydrodynamic diameter of 0.25 µM FUS progressively increased over time compared to the buffer-only condition (Supplementary Fig. 15a). At 5 mM ATP.$Mg^{2+}$, the mean cluster size initially increased compared to the buffer-only condition, then decreased and stabilized over time (Supplementary Fig. 15a). At 7.5 mM ATP.$Mg^{2+}$, the mean cluster size started at a smaller size and gradually increased, eventually stabilizing, resembling the 5 mM condition (Supplementary Fig. 15a). At 10 mM ATP.$Mg^{2+}$, DLS measurements were unable to detect cluster formation due to interference from the high ATP.$Mg^{2+}$ concentration.

The DLS data shown in Fig. 6a are consistent with FUS-SNAP observations. Lower ATP.$Mg^{2+}$ concentrations promote increased cluster size, while higher concentrations lead to size reduction. NTA data corroborated with DLS findings (Fig. 6b). Unlike DLS, at 10 mM ATP.$Mg^{2+}$, smaller clusters were detected in NTA experiments (Fig. 6b). Furthermore, the volume fraction of clusters increased at lower ATP.$Mg^{2+}$ concentrations and decreased at higher concentrations (Fig. 6c). These findings demonstrate that ATP.$Mg^{2+}$ actively modulates sub-saturation cluster formation in untagged FUS, exhibiting a concentration-dependent pattern that is both similar to and distinct from FUS-SNAP observations.

Next, we examined how NaXS and NaTS influence the clustering behavior of untagged FUS. As shown in Fig. 6d, clustering trends with NaXS resembled those with FUS-SNAP—lower concentrations increased cluster size, while higher concentrations decreased it. Specifically, 10 and 40 mM NaXS increased the mean cluster size of 0.25 µM FUS compared to buffer alone. Unlike FUS-SNAP, untagged FUS formed clusters even at 60 mM NaXS, but clustering was inhibited at 100 mM (Fig. 6d, Supplementary Fig. 15b). In contrast, 10 mM NaTS maintained cluster sizes similar to buffer, but increasing NaTS concentrations progressively reduced cluster size until full inhibition at 60 mM (Fig. 6d, Supplementary Fig. 15c). These results suggest NaXS and NaTS modulate FUS clustering in a concentration-dependent manner, with subtle differences at higher hydrotrope levels. NTA data supported these findings: cluster size distributions and volume fractions for 0.25 µM untagged FUS increased with NaXS up to 40 mM and then decreased at 100 mM (Fig. 6e, Supplementary

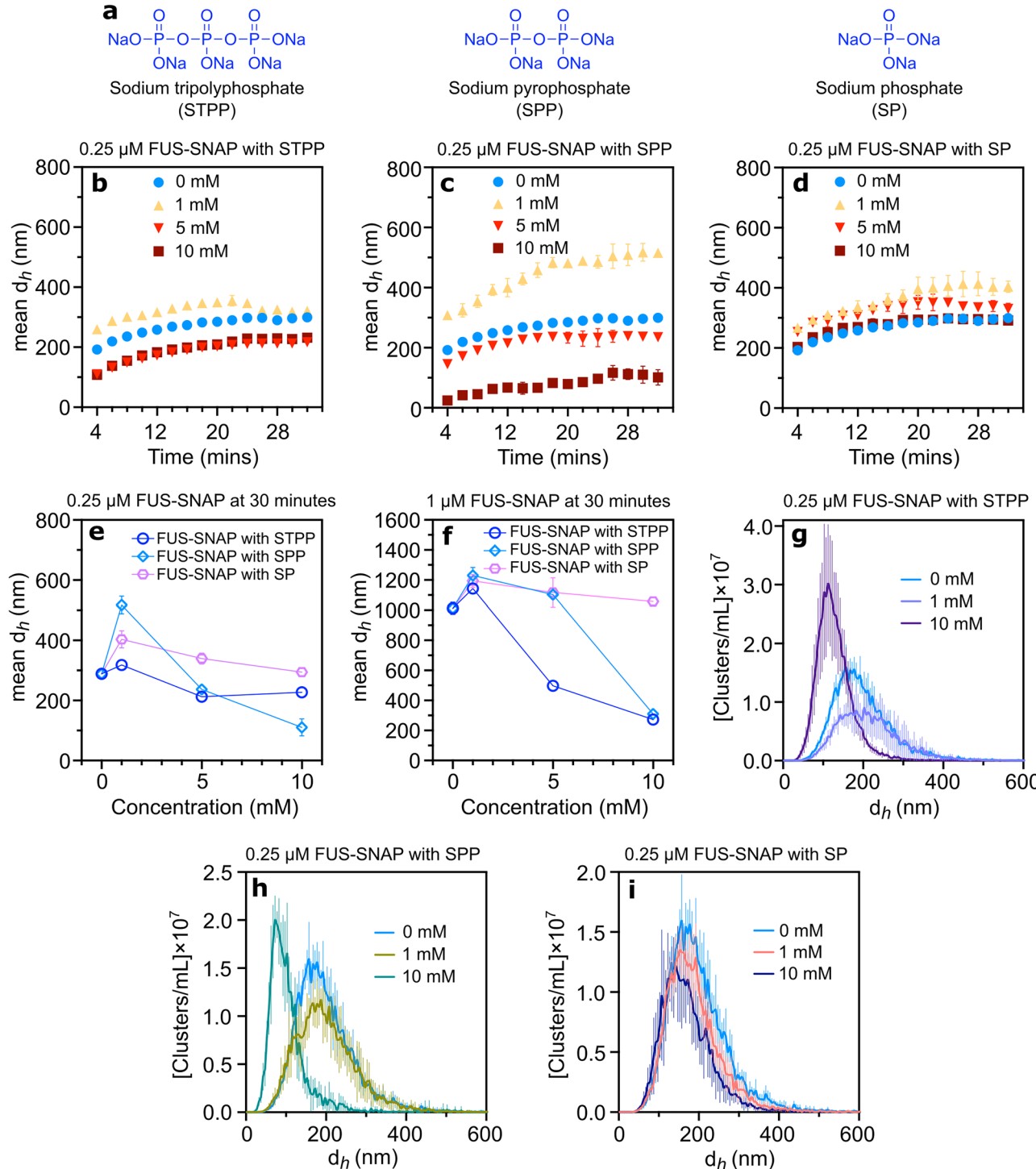

**Fig. 4 | Phosphates are salient in controlling the size of sub-saturated FUS-SNAP clusters.** Chemical structure of sodium tripolyphosphate (STPP), sodium pyrophosphate (SPP), and sodium phosphate (SP) (**a**). Dynamic Light Scattering (DLS) data show mesoscale clusters' mean hydrodynamic diameter ($d_h$) at 0.25 µM FUS-SNAP over 32 min with various concentrations of sodium tripolyphosphate (STPP) (**b**), sodium pyrophosphate (SPP) (**c**), and sodium phosphate (SP) (**d**). DLS data

show a mean $d_h$ of 0.25 µM (**e**) and 1 µM (**f**) FUS-SNAP at 30-min time points with various STPP, SPP, and SP concentrations. Nanoparticle Tracking Analysis (NTA) data at 0.25 µM FUS-SNAP for clusters' $d_h$ with STPP (**g**), SPP (**h**), and SP (**i**). NTA data are recorded at around 6–8 min from the time of sample preparation. Data are presented as mean ± SD from $n = 3$ (DLS) and $n = 5$ (NTA) independent experiments.

Fig. 16a). Similarly, NaTS concentrations showed comparable trends in NTA and DLS data (Fig. 6d, Supplementary Fig. 16b).

In the presence of 100 mM HD, the mean cluster size of 0.25 µM untagged FUS increased slightly compared to buffer-only conditions. As HD concentration increased, the cluster size of untagged FUS grew significantly (Fig. 6g, Supplementary Fig. 15d). Notably, unlike 0.25 µM FUS-

SNAP, untagged FUS showed a pronounced increase in cluster size at 500 mM and 1 M HD (Fig. 6g). Conversely, equivalent TEG concentrations caused only a slight increase in cluster size across all tested concentrations, mirroring the trend seen with FUS-SNAP (Fig. 6g, Supplementary Fig. 15e). NTA data supported these findings: cluster size distributions and volume fractions of 0.25 µM untagged FUS significantly increased with higher HD

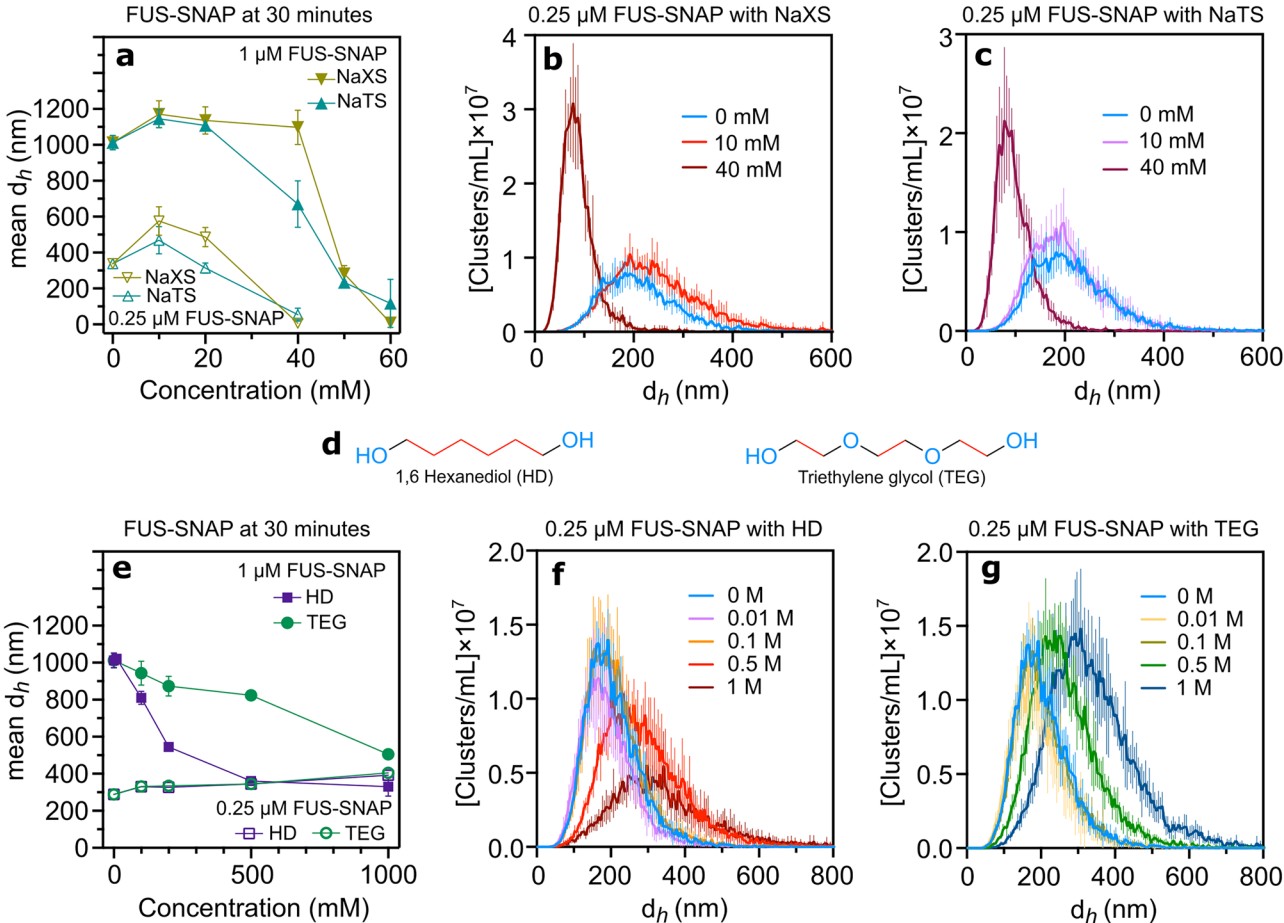

**Fig. 5 | Common hydrotropes—Sodium Xylene Sulfonate (NaXS), Sodium Toluene Sulfonate (NaTS), Hexanediol (HD), and Triethylene Glycol (TEG)— also influence the size of sub-saturation clusters.** Dynamic Light Scattering (DLS) data show mesoscale clusters' mean hydrodynamic diameter ($d_h$) of 0.25 µM and 1 µM FUS-SNAP at 30-min time points with various concentrations of Sodium Xylene sulfonate (NaXS) and Sodium Toluene Sulfonate (NaTS) (**a**). Nanoparticle Tracking Analysis (NTA) data at 0.25 µM FUS-SNAP for clusters' $d_h$ with NaXS (**b**) and NaTS (**c**); insets show corresponding volume fraction of clusters. Chemical structure of HD and TEG (**d**). DLS data show mesoscale clusters' mean $d_h$ of 0.25 µM and 1 µM FUS-SNAP at 30-minute time points with various concentrations of HD and TEG (**e**). NTA data at 0.25 µM FUS-SNAP for clusters' $d_h$ with HD (**f**) and TEG (**g**). NTA data are recorded at around 6–8 min from the time of sample preparation. Data are presented as mean ± SD from $n = 3$ (DLS) and $n = 5$ (NTA) independent experiments.

concentrations (Fig. 6h, Supplementary Fig. 16c). TEG also increased cluster size distributions and volume fractions compared to buffer-only, but the distributions were narrower than with HD (Fig. 6i, Supplementary Fig. 16d). While general clustering trends were similar for FUS-SNAP and untagged FUS with ATP, hydrotropes, HD, and TEG, notable differences under HD highlight the impact of the SNAP-tag on small-molecule interactions.

### The sequence composition of FET family proteins modulates their response to ATP and other small molecules, influencing sub-saturation clustering

FUS proteins are part of the FET family of RNA-binding proteins, known for their intrinsic disorder and RNA-binding domains. Despite structural similarities, differences in protein sequences and molecular weights give these proteins distinct chemical identities. To learn more, we examined the sub-saturation cluster formation of EWSR1-SNAP and TAF15-SNAP in the presence of ATP.Mg²⁺ and other small molecules. The SNAP-tagged versions were used for easier purification and handling because the tag increased the saturation concentration needed for phase separation.

### Effect of ATP.Mg²⁺ and hydrotropes on EWSR1-SNAP cluster formation

EWSR1-SNAP exhibited cluster formation trends comparable to FUS-SNAP in the presence of ATP.Mg²⁺ (Fig. 7a). At 1 mM ATP.Mg²⁺, the cluster size of 0.3 µM EWSR1-SNAP increased progressively over time

(Supplementary Fig. 16a). In contrast, at 5 mM and 10 mM ATP.Mg²⁺, the cluster size initially decreased and then reached a plateau over time, suggesting higher ATP.Mg²⁺ concentrations stabilize sub-saturation clusters (Supplementary Fig. 17a). NTA data confirmed these similar trends. At 1 mM ATP.Mg²⁺, cluster size distributions increased compared to buffer-only conditions (Fig. 7b). However, with 5 mM and 10 mM ATP.Mg²⁺, cluster size distributions decreased. The volume fraction of clusters of 0.3 µM EWSR1-SNAP increases at 1 mM ATP.Mg²⁺ and decreases with higher concentrations of ATP.Mg²⁺ (Fig. 7c).

In the presence of 10 mM NaXS and NaTS, the cluster size of EWSR1-SNAP increased compared to buffer-only conditions (Fig. 7d, Supplementary Fig. 17b, c). At 60 mM hydrotrope concentrations, cluster formation was inhibited. Similar to FUS-SNAP, NaTS had a more significant effect on reducing cluster size than NaXS, resulting in smaller clusters at equivalent concentrations. At 10 mM HD, the cluster size of EWSR1-SNAP increased compared to the buffer-only condition. However, at higher HD concentrations, the average cluster size remained similar but larger than in the buffer-only condition (Fig. 7d, Supplementary Fig 17d). TEG showed effects similar to HD, with comparable clustering behavior across concentrations (Fig. 7e, Supplementary Fig. 17e). These findings demonstrate that EWSR1-SNAP exhibits clustering patterns similar to FUS-SNAP in response to ATP.Mg²⁺ and small molecules, with consistent inhibition at high hydrotrope concentrations and stable cluster formation with HD and TEG.

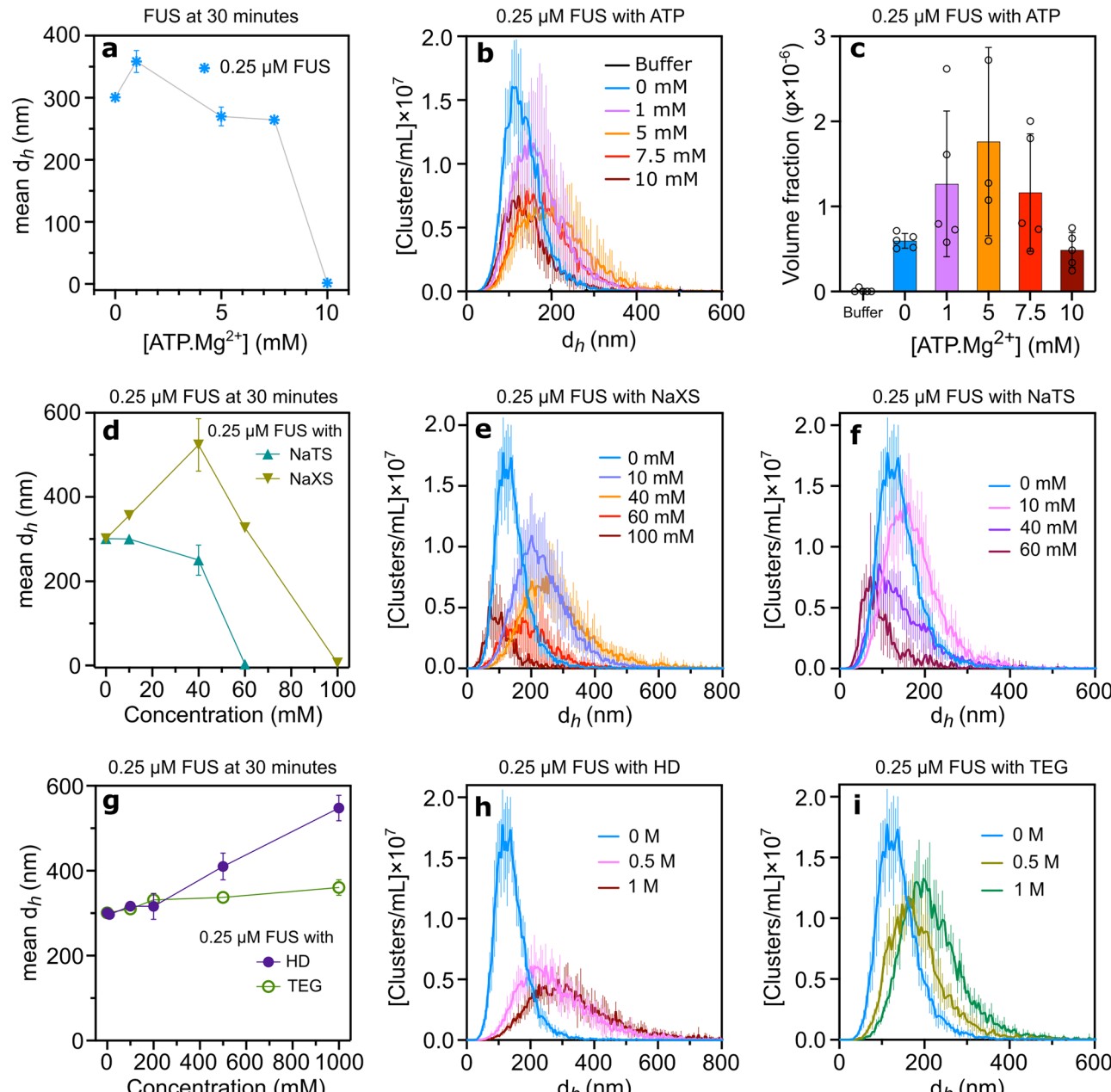

**Fig. 6 | Sub-saturation cluster formation of untagged FUS is similar to FUS-SNAP in the presence of ATP.Mg²⁺, hydrotopes (NaXS and NaTS), HD, and TEG.** Dynamic Light Scattering (DLS) data show mesoscale clusters' mean hydrodynamic diameter ($d_h$) of 0.25 μM FUS at 30-minute time points with various concentrations of ATP.Mg²⁺ (**a**). Nanoparticle Tracking Analysis (NTA) data for cluster hydrodynamic diameter at 0.25 μM FUS with different ATP.Mg²⁺ concentrations are shown in (**b**), while (**c**) displays the corresponding volume fraction of clusters. DLS data show mesoscale clusters' mean $d_h$ at 0.25 μM FUS at 30-min time points with various concentrations of NaTS and NaXS (**d**). NTA data at 0.25 μM FUS for clusters' $d_h$ with NaXS (**e**) and NaTS (**f**); insets show the corresponding volume fraction of clusters. DLS data show mesoscale clusters' mean $d_h$ at 0.25 μM FUS at 30-min time points with various concentrations of HD and TEG (**g**). NTA data at 0.25 μM FUS for clusters' $d_h$ with HD (**h**) and TEG (**i**); insets show the corresponding volume fraction of clusters. NTA data are recorded at around 6–8 min from the time of sample preparation. Data are presented as mean ± SD from $n$ = 3 (DLS) and $n$ = 5 (NTA) independent experiments.

## Effect of ATP.Mg²⁺ and hydrotopes on TAF15-SNAP cluster formation

Similar to FUS-SNAP and EWSR1-SNAP, TAF15-SNAP exhibits comparable trends in sub-saturation cluster formation in the presence of ATP.Mg²⁺ (Fig. 8a, Supplementary Fig. 18a). However, TAF15-SNAP requires higher ATP.Mg²⁺ concentrations to observe these trends compared to FUS-SNAP and EWSR1-SNAP. Notably, at 10 mM ATP.Mg²⁺, the cluster size of 0.25 μM TAF15-SNAP significantly increased over time compared to the buffer-only condition, unlike the stabilization observed with FUS-SNAP and EWSR1-SNAP (Fig. 7a). At higher ATP.Mg²⁺ concentrations (15 and

20 mM), cluster sizes decreased and stabilized over time (Supplementary Fig. 18a). NTA data corroborated these findings, showing that cluster size distributions initially increased with rising ATP.Mg²⁺ concentrations and then decreased at 20 mM ATP.Mg²⁺ (Fig. 8b). The volume fraction of clusters mirrored this trend (Fig. 8c).

In the presence of 10 mM NaXS and NaTS, the TAF15-SNAP cluster size increased compared to buffer-only conditions (Fig. 8d, Supplementary Fig. 18b, c). However, as hydrotrope concentrations increased to 40 mM, the cluster size decreased, and at 60 mM, cluster formation was completely inhibited. Interestingly, unlike FUS-SNAP and EWSR1-SNAP, NaXS had a

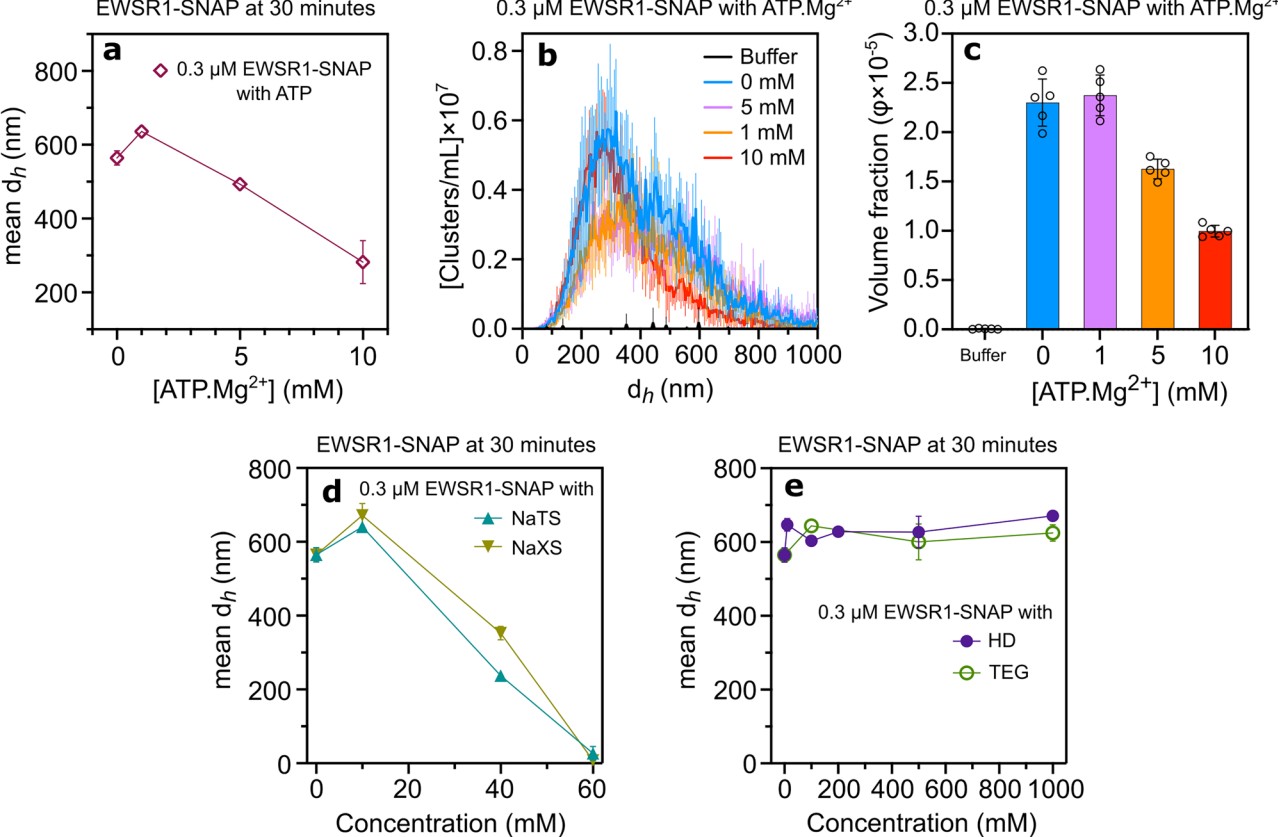

**Fig. 7 | ATP.Mg$^{2+}$, hydrotropes, and HD modulate the size of the Sub-saturation cluster of EWSR1-SNAP.** Dynamic Light Scattering (DLS) data show mesoscale clusters' mean hydrodynamic diameter ($d_h$) of 0.25 µM EWSR1-SNAP at 30-minute time points with various concentrations of APT.Mg$^{2+}$ (**a**). Nanoparticle Tracking Analysis (NTA) data for clusters' $d_h$ at 0.3 µM EWSR1-SNAP with different ATP.Mg$^{2+}$ concentrations are shown in (**b**), while (**c**) displays the corresponding volume fraction of clusters. DLS data show mesoscale clusters' mean $d_h$ at 30-minute time points with various NaTS and NaXS (**d**), HD, and TEG (**e**) concentrations. NTA data are recorded at around 6–8 min from sample preparation time. Data are presented as mean ± SD, with $n = 3$ (DLS) and $n = 5$ (NTA) independent samples.

more substantial inhibitory effect on TAF15-SNAP cluster formation than NaTS, resulting in smaller mean cluster sizes. In the presence of HD, the cluster size of 0.25 µM TAF15-SNAP increased progressively with rising HD concentrations, peaking at 500 mM, where the size was approximately two-fold higher compared to buffer-only conditions (Fig. 8e, Supplementary Fig. 18d). At 1000 mM (1 M) HD, cluster size decreased compared to the 500 mM condition. Conversely, TEG caused a gradual increase in cluster size with increasing concentrations, albeit less pronounced than HD (Fig. 8e, Supplementary Fig. 18e).

**Depending on the inherent molecular chemistry, ATP.Mg$^{2+}$, NaXS, NaTS, and HD interact differently with FUS-SNAP**

Here, we examined the thermal stability of FUS-SNAP in the presence of ATP, hydrotropes, and HD. We utilized nanoscale differential scanning fluorimetry (NanoDSF), a modified method for assessing protein stability. This technique measures intrinsic tryptophan and tyrosine fluorescence by monitoring two wavelengths, 330 nm and 350 nm, to analyze the protein's thermal unfolding. For the NanoDSF experiment, we used 1 µM FUS-SNAP with various concentrations of ATP.Mg$^{2+}$, NaXS, NaTS, and HD.

Figure 9 shows that, under buffer-only conditions, the transition temperature ($T_m$) of FUS-SNAP is ~51.9 °C. The temperature-induced denaturation pattern exhibits a blueshift or solvent burial (Supplementary Fig. 19a). In the presence of 1 mM ATP.Mg$^{2+}$, we observe a similar $T_m$ value, around 51.3 °C. At 5 mM ATP.Mg$^{2+}$, the $T_m$ increased to about 54.2 °C, while at 10 mM ATP.Mg$^{2+}$, it decreased to roughly 48.5 °C. When ATP.Mg$^{2+}$ is present, protein denaturation also shows a blueshift or solvent burial (Fig. 9a). With 10 mM NaXS, the transition temperature increased to ~63 °C, accompanied by a redshift during unfolding. This suggests that

tryptophan (Trp) residues, initially buried within the protein's hydrophobic core, become exposed to the solvent during unfolding (Supplementary Fig. 19b). At 60 mM NaXS, no transition temperature was observed, as no signal (ratio of 330/350) was detected (Fig. 9b). With NaTS, both blue and redshift or solvent-exposed unfolding were recorded at 49.7 °C and 64.4 °C with 10 mM NaTS, and at 49.5 °C and 65.4 °C with 60 mM NaTS (Fig. 9c). In the presence of HD, the Tm decreased to 51 °C, 50.4 °C, and 49.3 °C at 10 mM, 100 mM, and 200 mM HD, respectively, displaying a blueshift pattern similar to the buffer-only condition (Fig. 9d). These results suggest that interactions between FUS-SNAP and small amphiphilic molecules differ significantly due to their chemical properties.

These findings highlight that sequence-specific chemical properties play a critical role in determining the effects of small molecules on sub-saturation cluster formation in EWSR1-SNAP and TAF15-SNAP. Although common regulatory patterns emerge across FET family proteins, differences in response to ATP.Mg$^{2+}$, hydrotropes, and HD underscore the significance of protein-specific sequence and structural elements.

## Discussion

Previous studies have demonstrated that increasing concentrations of ATP.Mg$^{2+}$ lead to the dissolution of phase-separated droplets of FET proteins, with ATP also characterized as a 'biological hydrotrope' due to its structural and functional similarity to hydrotropes and its role in modulating protein phase behavior[19]. At lower concentrations (1–2 mM), ATP enhances phase separation, but at higher concentrations (>6 mM), it dissolves phase-separated condensates[29]. Additionally, ATP.Mg$^{2+}$ inhibits the coarsening of phase separation and stabilizes sub-saturation clusters of FET proteins at saturation concentration[9]. In this study, we demonstrated that

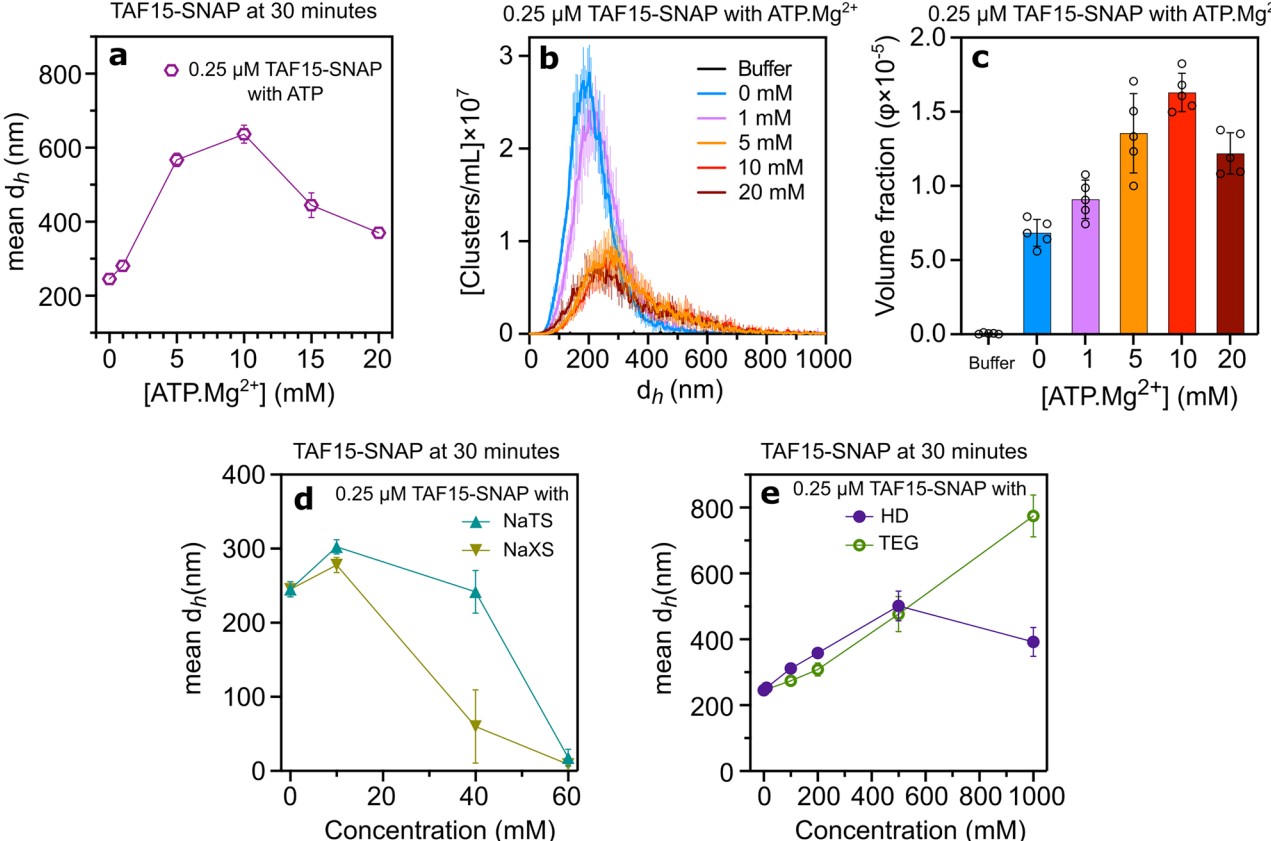

**Fig. 8 | ATP.Mg$^{2+}$, hydrotropes, HD, and TEG modulate the size of the Sub-saturation cluster of TAF15-SNAP.** Dynamic Light Scattering (DLS) data show mesoscale clusters' mean hydrodynamic diameter ($d_h$) of 0.25 μM TAF15-SNAP at 30-minute time points with various concentrations of APT.Mg$^{2+}$ (**a**). Nanoparticle Tracking Analysis (NTA) data for clusters' $d_h$ at 0.25 μM TAF15-SNAP with different ATP.Mg$^{2+}$ concentrations are shown in (**b**), while (**c**) displays the corresponding volume fraction of clusters. DLS data show mesoscale clusters' mean $d_h$ at 30-minute time points with various NaTS and NaXS (**d**), HD, and TEG (**e**) concentrations. NTA data are recorded at around 6–8 min from sample preparation time. Data are presented as mean ± SD, with $n = 3$ (DLS) and $n = 5$ (NTA) independent samples.

ATP modulates the size of mesoscale clusters in sub-saturation concentrations of FET family proteins. Cluster size increased at lower ATP levels but decreased at higher concentrations, a behavior that cannot be fully explained by hydrotropic effects. Importantly, these clusters persisted even at physiologically relevant high ATP concentrations. The enhancement of phase separation at low ATP levels is attributed to ATP acting as a bivalent binder through non-specific interactions: the adenine ring interacts with aromatic residues via π-π or π-cation interactions[22,34], and the triphosphate chain forms electrostatic interactions with arginine and lysine residues[29]. Similar effects may be attributed to the enhancement of the size of mesoscale clusters at low concentrations of ATP.Mg$^{2+}$ at sub-saturation concentrations of FET proteins. Furthermore, Mehringer et al. suggest that ATP is not a classic hydrotrope but a kosmotropic ion according to the Hofmeister series[35]. When comparing the effects of ATP.Mg$^{2+}$, ADP, AMP, and phosphates, we observed similar trends, indicating they influence clusters via analogous mechanisms. Kosmotropic ions interact with water, leading to 'salting-in' or 'salting-out' effects[36], while leaving out the chemical details of the surfaces of proteins[36,37]. Higher concentrations of nucleotides and phosphates inhibit clustering by dissolving proteins, but the number of phosphate groups does not consistently correlate with clustering effects. At lower concentrations of nucleotides and phosphates, the kosmotropic effect alone cannot explain the increased cluster size, suggesting non-covalent interactions play a key role.

Previous studies have shown that arginine–phosphate interactions influence the phase behavior of disordered peptides[38]. At low concentrations, these interactions act as mild crosslinkers, increasing cluster size, but become saturated at higher concentrations, leading to stabilization of monomers and clusters. NMR studies reveal interactions involving arginine-rich domains, lysine, and other residues in proteins like FUS and TDP-43[39–41], as well as non-specific interactions between ATP and lysozyme[42]. This study emphasizes clustering at sub-saturation protein concentrations in the presence of ATP, where similar interactions likely contribute, especially arginine–phosphate interactions, given the arginine content in FET proteins such as FUS, EWSR1, and TAF15, which correlate with their clustering behavior. TAF15, with the highest arginine content, requires higher ATP·Mg$^{2+}$ concentrations to reduce cluster sizes. Additionally, ATP·Mg$^{2+}$ interactions with FUS, examined at different pH levels, show that ATP modulates clustering differently below and above its pKa (~6.5). At pH 5.5, phosphate groups are predominantly protonated, reducing interactions between FUS-SNAP and ATP. In contrast, at pH levels of 7.4 and 8.5, phosphate groups can interact with arginine and lysine residues, enabling strong modulation of cluster size by ATP across varying concentrations.

Hydrotropes like NaXS and NaTS influence FET proteins clustering similarly to ATP·Mg$^{2+}$ but require about ten times higher concentrations to produce comparable effects. This is because proteins possess conserved phosphate-binding sites[43], and ATP phosphate has two negative charges, leading to stronger binding affinity. NaXS and NaTS carry a single negative charge at physiological pH, and their hydrophobic xylene and toluene groups may engage in π-π and hydrophobic interactions with proteins. At low concentrations, they act as crosslinkers, promoting clusters, whereas at higher concentrations, they saturate binding sites, inhibiting clustering. Structural differences between NaXS and NaTS affect their regulatory roles too. NaXS, with two methyl groups, is more hydrophobic than NaTS, which

**Fig. 9 | Interactions with ATP and other small amphiphilic molecules alter the transition temperature of FUS-SNAP probed by NanoDSF.** 1st Derivatives of the unfolding curves (350/330) ratio with the apparent transition temperatures of FUS-SNAP with ATP.Mg$^{2+}$ (**a**), NaXS (**b**), NaTS (**c**), and HD (**d**). Data are presented as mean ± SD, with $n = 3$ independent samples.

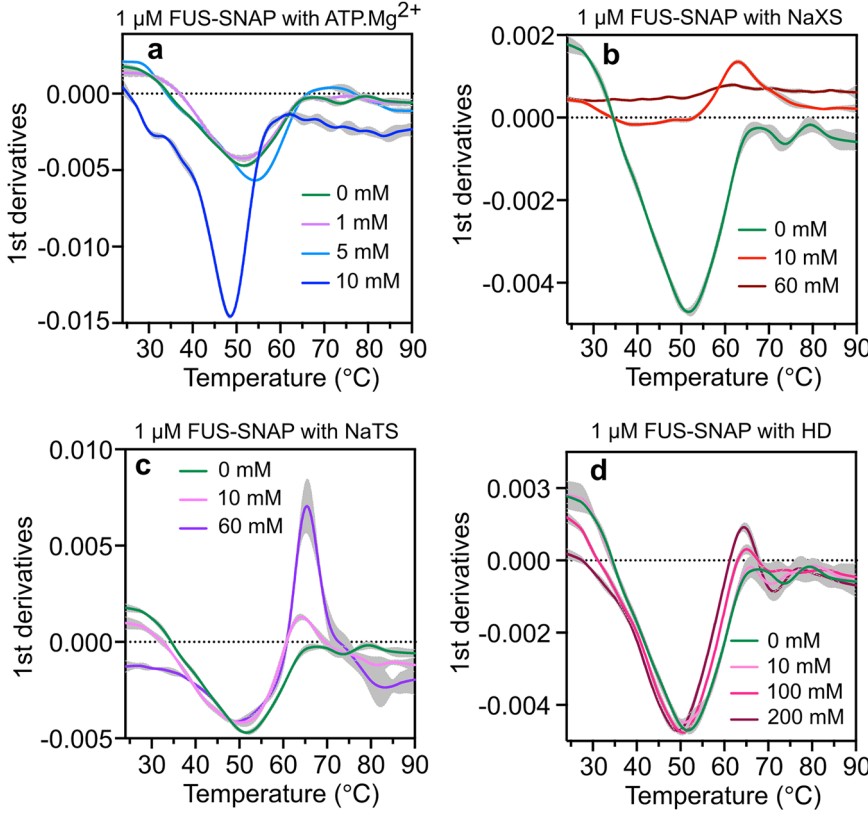

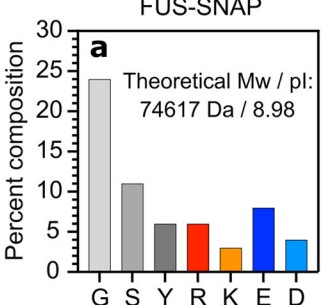 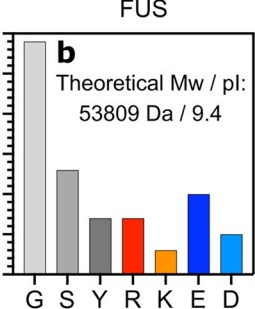 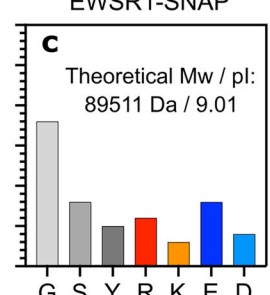 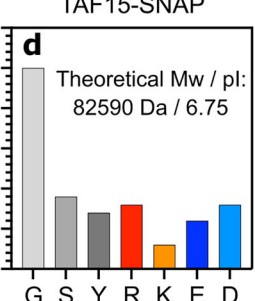

**Fig. 10 | Sequence composition of FET proteins.** The percent composition of cationic amino acids—arginine (R) and lysine (K); anionic amino acids—glutamic acid (E) and aspartic acid (D); and the most abundant amino acids—glycine (G), serine (S), and tyrosine (Y) is shown for FUS-SNAP (**a**), FUS (**b**), EWSR1-SNAP (**c**), and TAF15-SNAP (**d**). The theoretical molecular weight (Mw) and isoelectric point (pI) are reported for each protein.

has only one. The sequence diversity among FET proteins like FUS, EWSR1, and TAF15, especially variations in their isoelectric points (Fig. 10), influences how these hydrotropes modulate clustering. HD, containing hydrophobic methylene groups and weak hydrophilic sites, interacts with hydrophobic regions, stabilizing proteins and preventing coarsening. TEG, with similar carbon content but more oxygen atoms, exhibits milder effects due to reduced hydrophobicity. Furthermore, NanoDSF analysis shows that small molecules affect the thermal stability of FUS-SNAP, suggesting similar mechanisms. These interactions are modulated by protein sequences and the inherent chemistry of the small molecules.

Three concentration-dependent scenarios are observed for sub-saturation clusters: at low levels, ATP, hydrotropes, and HD act as mild crosslinkers, increasing cluster size; at moderate levels, they stabilize clusters by organizing amphiphilic molecules to prevent growth; and at high levels, they inhibit further clustering, but smaller mesoscale clusters persist.

The non-monotonic phase behavior with increasing concentration of an additional component resembles that in polymer systems with a cosolvent that interacts preferentially with polymers or proteins[44–46]. As cosolvent levels rise, polymer condensation occurs, followed by re-entry at higher concentrations. Simulation results support predictions from the adsorption-attraction mean-field theory[47,48], which explains this behavior by a collective transition driven by transient monomer-cosolvent bridging. This mechanism, previously called *gluonic*[49], involves nonspecific attractive interactions facilitated by weak, transient bonds. The present study indicates biological hydrotropes like ATP, NaXS, NaTS, HD, and TEG act as gluonic components, promoting FET protein clustering.

## Conclusion

In cellular environments, ATP concentrations vary across tissues, from 2.88 mM in the brain to 7.47 mM in cardiac muscle[18], suggesting that ATP may differentially influence FET protein clustering depending on cell type. This study reveals that ATP and small amphiphilic molecules regulate FET protein clusters in a concentration-dependent manner. These findings challenge traditional hydrotropic and kosmotropic models, emphasizing

the importance of sequence-specific interactions between proteins and small molecules in cluster regulation. Given that FET proteins are involved in pathogenic aggregation in neurodegenerative diseases like FTLD and ALS[8]—with lower ATP levels in the brain potentially promoting aggregation—these insights could lead to novel strategies for regulating protein assemblies, both in healthy and diseased states.

## Methods

### Protein purification

Protein purification was performed according to established protocols[9,10]. Recombinant proteins were expressed in SF9 insect cells maintained in suspension at 27 °C using ESF921 serum-free medium. The FlexiBAC system facilitated baculovirus production, enabling target gene expression under the control of the polyhedrin promoter. In a typical experiment, 1 L of SF9 cells at a density of 1 million cells per mL were infected with 5 mL of P2 virus and harvested 72 h post-infection. Cells were collected by centrifugation at 300 RCF for 15 min at 4 °C, and the pellet was resuspended in 30 mL of ice-cold lysis buffer (50 mM HEPES pH 7.4, 1 M KCl, and 5% glycerol) supplemented with EDTA-free protease inhibitor tablets. Cell disruption was achieved by sonication on ice for 5 min at 35% output with a 50% duty cycle. Cellular debris was removed by centrifugation at 39,800 RCF for 30 min at 4 °C. Subsequent purification steps were performed at room temperature. The cleared lysate was passed through a 5 mL Ni-NTA agarose column (Protino, Macherey-Nagel) using a peristaltic pump. The column was washed with 10 column volumes of lysis buffer containing 10 mM imidazole. Bound protein was eluted with 5 column volumes of elution buffer (50 mM HEPES pH 7.4, 1 M KCl, 5% glycerol, and 300 mM imidazole). Peak fractions were incubated with 10 mL of amylose resin for 10 min, followed by gravity flow drainage. The resin was washed with 10 column volumes of lysis buffer, and the target protein was eluted using 5 column volumes of maltose-containing buffer (50 mM HEPES pH 7.4, 1 M KCl, 5% glycerol, and 30 mM maltose). Protein concentration was monitored using a Bradford assay, and purity was assessed by SDS-PAGE. To remove the N-terminal His-MBP tag, 3C protease was added at a molar ratio of 1:100, and digestion was carried out at room temperature for 4 h. The sample was further purified using size-exclusion chromatography on an ÄKTA system (GE Healthcare) with a Superdex 200 10/300 Increase column pre-equilibrated with storage buffer (50 mM HEPES pH 7.4, 500 mM KCl, 5% glycerol, and 1 mM DTT). Peak fractions of C-terminal SNAP-tagged proteins were collected, concentrated, and prepared for experiments. For untagged protein preparation, TEV protease was added at a 1:50 molar ratio, and the mixture was incubated at room temperature for 6 h. Gel filtration chromatography purified the resulting protein using a Superdex 200 10/300 Increase column equilibrated with storage buffer. Peak fractions were pooled and concentrated using a 30 kDa molecular weight cut-off (MWCO) filter at 3000 RCF at room temperature. Protein concentration was determined using a NanoDrop ND-1000 spectrophotometer (Thermo Scientific), with 260/280 ratios ranging from 0.52 to 0.56. Peak fractions were pooled and immediately used for dynamic light scattering (DLS) or nanoparticle tracking analysis (NTA) experiments. SNAP-tagged and untagged proteins were snap-frozen in liquid nitrogen and stored at −80 °C.

### Preparation of ATP.Mg$^{2+}$

Following the previous report[19], nucleotide-magnesium complexes were prepared. Briefly, molar equivalent magnesium acetate was added to the solutions of adenosine triphosphate sodium salt and adenosine diphosphate sodium salt and incubated for 5 min before the experiments.

### DLS measurements

DLS measurements were carried out at 24 °C using a Wyatt DynaPro® Nanostar (Waters|Wyatt Technology, US) instrument. Disposable cuvettes (Uvette 220-1600 nm) from Eppendorf were employed, with a sample volume of 100 µl. A 658 nm laser illuminated the sample solutions, and the intensity of light scattered at an angle of 90° was measured using a single photon counting module. This allows for the measurement of time-

dependent fluctuations in the intensity of scattered light as scatterers undergo Brownian motion. The rate of fluctuations is directly related to the diffusion rate of the molecule through the solvent, which in turn correlates with the particles' hydrodynamic radii. Analyzing intensity fluctuations enables the determination of the diffusion coefficients ($D_t$) of particles, which are converted into a size distribution using the Stokes-Einstein equation (Eq. (1)).

Stokes–Einstein relation:

$$D_t = \frac{k_B T}{6\pi\eta R_h} \tag{1}$$

Here, $k_B$ is the Boltzmann constant ($1.381 \times 10^{-23}$ J/K) and η is the absolute (or dynamic) viscosity of the solvent. In this work, we used the hydrodynamic diameter $d_h$ (i.e., $d_h = 2R_h$) as the preferred way to quantify particle sizes.

The analyte's translational diffusion coefficient ($D_t$) is obtained by automated nonlinear least squares fitting of the autocorrelation function that quantitatively describes the measured time-dependent fluctuations in light scattering intensity. The analysis is done directly using the accompanying DYNAMICS™ software.

For the measurements, all buffer solutions were filtered using 0.2 µm membranes (Millex®-GS units) purchased from Millipore™. All protein samples were centrifuged at 10,000 RCF before measurements. All experiments were conducted with the following settings: Material – protein; Dispersant – buffers; Temperature: 24 °C with equilibration time – 120 s, Measurement angle: 90°. Each spectrum represents the average of 12 scans, each of 10 seconds in duration. The samples were prepared by adding centrifuged stock proteins, followed by dilution buffer, and mixed thoroughly by pipetting three to four times. The samples were equilibrated for 2 min at 24 °C, and the data were recorded in 2-min intervals. DYNAMICS™ (Waters|Wyatt Technology, USA) software is used to control experimental parameters, collect data, and analyze it. The autocorrelation, intensity size distributions, and mean hydrodynamic diameters of each measurement were exported from DYNAMICS™ software and plotted in GraphPad Prism software.

In this report, we present only the intensity size distribution, as it directly correlates with the autocorrelation function without requiring any data transformation, as FET proteins' mesoscale clusters are polydisperse and quasi-spherical[9]. For volume transformation, assuming spherical particles, the particle volume is proportional to the cube of the size. For number transformation, assuming isotropic small particles, the scattering intensity from a spherical particle is proportional to the sixth power of the size.

Note: The proteins were purified and stored in a high-salt (500 mM KCl) buffer, which contributes background salt to the final KCl concentration in the experiments. For FUS-SNAP, the stock solution can be concentrated to ~50 µM, providing a broader range of concentrations for experiments. However, for untagged FUS, EWSR1-SNAP, and TAF15-SNAP, the stock concentrations remain below 20 µM. To maintain a final KCl concentration of ~10 mM under sub-saturation conditions, we use lower protein concentrations in experiments, such as 0.25 µM for untagged FUS and TAF15-SNAP, and 0.3 µM for EWSR1-SNAP.

### Nanoparticle Tracking Analysis (NTA)

Nanoparticle tracking analysis was performed using the NanoSight Pro from Malvern Instruments, which has a measurement range of 10 nm to 1 µm. The system included a NanoSight syringe pump to inject samples for the experiments. NTA measurements utilize the properties of light scattering and Brownian motion to determine the size distributions and concentrations of particles in liquid suspension. A laser beam (488 nm) was directed through the sample chamber, and the suspended particles were visualized using a 20× magnification microscope. A video recording capturing the movement of particles under Brownian motion was taken with a camera mounted on the microscope, operating at 30 frames per second. The software tracks the movement of individual particles from frame to frame to

calculate the mean square displacement. The Stokes-Einstein equation (Eq. (1)) is used to determine the particles' hydrodynamic diameter. The instrument provides the particle concentration by counting particles per frame within a known observation volume.

All buffers were filtered through a 0.22 µm polyvinylidene fluoride membrane filter (Merck, Germany). Protein stock solutions were centrifuged at 20000 RCF for 5 min at room temperature, and the concentration was measured before further measurements. Samples were prepared by adding the centrifuged stock proteins, followed by a dilution buffer, and mixed thoroughly by pipetting 4–6 times. The samples were allowed to equilibrate for 2 min at 24 °C, and data were recorded ~6 min after sample preparation.

### Calculation of volume fraction

Nanoparticle Tracking Analysis (NTA) provides particles' hydrodynamic diameters along with their concentration in particles per milliliter (particles/mL).

$$\text{The volume fraction of clusters } (\phi) \text{ is calculated as}: \frac{\Sigma \; volume \; of \; particles}{Total \; volume \; (1 \; mL)}$$

### Nano Differential Scanning Fluorimetry (nanoDSF)

The thermal unfolding of FUS-SNAP was carried out using nanoDSF with a Prometheus NT.48 (NanoTemper Technologies, Munich, Germany) instrument. Protein samples were prepared in buffers and briefly centrifuged (5 min, 10,000 RCF) to eliminate protein aggregates. Samples were loaded into high-sensitivity glass capillaries (Cat#PR-C006, NanoTemper Technologies, Munich, Germany) and subjected to a linear thermal ramp from 20 °C to 95 °C at a rate of 1 °C/min. Intrinsic protein fluorescence emission was collected at 330 and 350 nm with a dual-UV detector over a temperature gradient. The fluorescence intensity ratio (350/330) was plotted against temperature, and the inflection point of the transition was obtained from the maximum of the first derivative for each measurement using Therm-Control Software (NanoTemper Technologies, Munich, Germany). All experiments were conducted in triplicate; the mean and standard deviation were calculated for all three measurements.

### Reporting summary

Further information on research design is available in the Nature Portfolio Reporting Summary linked to this article.

### Data availability

All raw data for the main text and/or Supplementary Figs. are available for free download at: https://doi.org/10.6084/m9.figshare.30375283.

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

## Acknowledgements
I sincerely thank Albena Lederer, Anthony A. Hyman, Jens-Uwe Sommer, Carsten Werner, Susanne Boye, and Tyler Harmon for their valuable discussions and insights. I am also grateful to the Hyman Laboratory at the Max Planck Institute for Molecular Cell Biology and Genetics (MPI-CBG) for providing the DNA constructs for all proteins and to the PEPC facility at MPI-CBG for their support with protein expression, purification, and characterization. This work was funded by Deutsche Forschungsgemeinschaft (DFG —562384767).

## Author contributions
Conceptualization, experiment design, analysis, writing, editing, revising, and funding acquisition: M.K.

## Funding

## Competing interests
The author declares no competing interests.
