## [Transparent Peer Review file · Communications Chemistry]

ATP and Small Amphiphilic Molecules Act as Molecular Matchmakers to Fine-Tune FET Protein Clusters

Corresponding Author: Dr Mrityunjy Kar

Version 0:

Reviewer comments:

Reviewer #1

(Remarks to the Author)

The manuscript describes the effect of varying concentrations of the physiological metabolite ATP on the molecular assembly of FET proteins in vitro. The focus of the work is the formation of mesoscale molecular assemblies (clusters), which form at concentrations well below the saturation threshold for phase separation. This protein family has been associated with neurodegenerative disorders, thereby understanding the effect of metabolites in their aggregation propensity has biomedical relevance. The biophysical study was performed via dynamic light scattering (DLS) and nanoparticle tracking analysis (NTA) using ATP, ADP, AMP and other small amphiphilic molecules to dissect the role of the different chemical groups on the aggregation propensity. FUS-SNAP data were compared with untagged FUS and FUS tagged with other groups to characterize the additional effect of the tag on the aggregation behaviour. The data obtained with all cosolutes and all FUS variants were systematically compared to draw the conclusions (with an appropriate statistical analysis). NanoDSF was also used to characterize the binding of cosolutes to the proteins via changes in thermal protein's stability. The author finds that the phosphate moiety in nucleotides plays a pivotal role in regulating the size of FUS-SNAP clusters and that sequence-specific chemical properties of all cosolute tested play a critical role in determining the effects of small molecules on sub-saturation cluster formation. A complex picture emerges, which highlights the roles of different chemical interactions in the overall clustering response of the FET family. The author suggests three possible scenarios related to the physiological role of ATP, which helps summarizing the large amount of data collected and proposes that sequence-specific interactions between the protein and ATP modulate the cluster formation, challenging conventional hydrotropic and kosmotropic models of action and possibly fostering new studies on the clustering of FET proteins. The manuscript is interesting and timely, and proposes a novel idea on the modulation of the clustering of FET proteins, which is interesting for a general audience.

Below, two minor points which could be addressed:

- 1) Besides the average concentrations in different cells, it would be interesting to discuss some known changes of ATP levels in the same cell during cellular homeostasis, which could modulate the formation of the assemblies.
- 2) The effect of pH on the ATP-induced molecular assemblies was not characterized, and the possible role of changes in the local pH in cellular milieu as additional regulator of the aggregation should be mentioned.

Reviewer #2

(Remarks to the Author)

FET (FUS-EWSR1-TAF15) family proteins have the ability to undergo liquid-liquid phase separation (LLPS). Previous studies have shown that FET proteins inherently form mesoscale molecular assemblies, known as clusters, even at concentrations well below the threshold for LLPS. Given the critical role of FET proteins in neurodegenerative diseases, there is growing interest in understanding their LLPS behaviour and cluster formation mechanisms.

This study demonstrates that adenosine triphosphate (ATP)—an amphiphilic molecule and essential cellular metabolite—modulates the size of these sub-saturation mesoscale clusters in a concentration-dependent manner. At lower ATP concentrations, cluster sizes increase, while higher ATP levels lead to reduced cluster size. Other amphiphilic molecules, including common hydrotropes such as sodium xylene sulfonate and sodium toluene sulfonate, exhibit similar concentration-dependent effects on FET protein clustering.

All experiments in this study were conducted under non-physiological conditions and in most cases, proteins had tag. As a result, the biological significance of these findings remains unclear. There are a number of points that need to be addressed by the authors.

Major Concerns:

1. Non-physiological Ionic Strength: All experiments in this study were conducted at low ionic strength (10 mM KCl), which is significantly below physiological levels. The authors state that higher KCl concentrations inhibit FET protein clustering. However, it remains unclear whether physiological salt concentrations merely inhibit clustering or completely abrogate it. If clustering is abrogated under physiological conditions, the relevance of these findings becomes questionable. If only inhibited, to what extent does this occur? What is the effect of ATP and other amphiphilic small molecules on sub-saturation cluster formation under physiological KCl concentrations? This information is critical for assessing the physiological significance of the study.
2. Statistical Significance and Data Interpretation: The manuscript frequently describes observed changes as "slight size increase" (e.g., lines 173–174), without providing statistical analysis to support these claims. Wherever possible, the authors should include statistical significance testing. Specifically, Figures 2 through 8 should include appropriate statistical analyses (e.g., p-values), and the interpretation of results in these figures should be revisited in light of the statistical outcomes.
3. Impact of SNAP-Tag on Cluster Formation in the presence of HD: The author established that the SNAP-tag can significantly influence FUS cluster formation, particularly in relation to the effects of 1,6-hexanediol (HD). The author demonstrated that the effects of HD are similar for FUS-SNAP and EWSR1-SNAP but differ for TAF15-SNAP. Given the known impact of SNAP-tag on protein behaviour, HD-related experiments should be repeated using untagged proteins to accurately assess the role of HD in cluster formation across the FET family.
4. Use of Tagged Proteins. The protein concentrations used in Figures 2–8 is very low (typically 0.25 μ M), and the sample volumes are small. Given the potential for the SNAP-tag to affect formation of clusters, I would recommend that dynamic light scattering (DLS) or nanoparticle tracking analysis (NTA) experiments—especially those involving ATP, physiologically relevant molecule tested—be conducted using untagged proteins.

Minor Concerns:

1. Lines 141 and 146. The manuscript refers to Figure 2f in both instances, but it should be corrected to Figure 2d.
2. Line 173. The phrase "obscure data" is unclear—please clarify its meaning. Additionally, why are the data for 5 and 10 mM AMP missing in the DLS results (Figures 3c and 3e), while the corresponding NTA data are shown in Figure 3h?
3. In Figure 7, the author uses 0.3 μ M EWSR1-SNAP, whereas 0.25 μ M protein concentrations are used elsewhere (e.g., for FUS-SNAP, FUS, and TAF15-SNAP). The rationale for using a different concentration for EWSR1-SNAP should be clarified, as this inconsistency complicates direct comparison across proteins.
4. The volume fraction of clusters for both FUS-SNAP and EWSR1-SNAP increases at 1 mM ATP and decreases at 5 mM ATP, indicating a similar response. However, EWSR1-SNAP contains 10 more arginine residues than FUS-SNAP. Conversely, TAF15-SNAP also contains 10 more arginines than EWSR1-SNAP but requires significantly higher ATP concentrations (15–20 mM) to observe a similar decrease in cluster size. How does author reconcile the similar ATP responses between FUS-SNAP and EWSR1-SNAP despite their arginine content differences, while TAF15-SNAP, with a comparable arginine difference relative to EWSR1-SNAP, shows a markedly different behaviour?

Reviewer #3

(Remarks to the Author)

The author of "Molecular Matchmakers: How ATP and Small Amphiphilic Molecules Fine-Tune FET Protein Clusters" explores how nucleotides with varying amounts of phosphate as well as hydrotropes and the amphiphilic molecule 1,6-hexanediol influence the propensity of sub-micron protein clusters of FET proteins to form. With the use of dynamic light scattering and nanoparticle tracking analysis the size of clusters is monitored over time for each solute, showing that clusters generally grow in size and saturate around 30 minutes. Cluster formation depends on the solute concentration, where increased solute decreases cluster size or inhibits cluster formation completely through disruption of pi-pi and electrostatic interactions. The author concludes that this behavior comes from sequence specific interactions.

There is potential for this work to be of broad interest, but the interpretation of the data as well as the conclusions are the same as the previously published paper Patel, Science, 2017, that investigated the hydrotrope effect of many of the same molecules as in this work. The findings here do not provide new insight into the molecular mechanism compared to the previous literature. The main new point of the work is that the hydrotrope effect is observed at the cluster sub micron size, compared to the mesoscopic super micron condensate, which is not surprising. It is recommended that major revisions are made to the manuscript

Major concerns:

- Three techniques are used throughout the paper, two of which track particle size. These methods provide no insight into molecular mechanisms of ATP-protein interactions. The manuscript would be strengthened through the addition of complementary spectroscopic measurements, such as NMR or EPR, or vibrational techniques such as two dimensional infrared, that provide information on structure and dynamics of protein complexes. This would especially shed light onto what sequence specific interactions are important for the regulation of cluster size
- Data provided in the manuscript is comprehensive, but to the point that it is too dense. Figures are small and often hard to read, especially those with insets, and panels within a figure often have repetitive data. Shifting some data to the supporting

information and increasing figure size would improve accessibility of the data.

- For some data sets, concentrations are missing or are reported as inconclusive and left out.
- It is concluded in the manuscript that ATP is not a kosmotrope due to its concentration dependent behavior. However, it seems the behavior observed here is consistent with kosmotropes as they often exhibit nonlinear behavior with concentration where the protein is stabilized at low concentration and is destabilized at high concentrations. Additionally, many kosmotropes/chaotropes have non-covalent interactions with other molecules through solvent shared ion pairs or contact-ion pairs, and their behavior is unique to the specific surface chemistry of the protein.

Minor Concerns

- The number of times data were reproduced is not mentioned in the manuscript
- Autocorrelation functions of DLS with no slow modes are referred to throughout the text, but how this conclusion is made from the plotted data is unclear.
- Error of the melting temperature is not reported
- Discussion of redshift and blueshift of NanoDSF data in terms of solvent exposed/ solvent buried would make the conclusions more accessible to a broader audience

Version 1:

Reviewer comments:

Reviewer #1

(Remarks to the Author)

The author has addressed my concerns in the revised version of the manuscript. The additional data on the pH dependency of the ATP-induced molecular assemblies provide further insights into the process of molecular assembly of FET proteins in vitro, and shed light on the role of the phosphate groups of ATP in the process. The author agrees that increasing the complexity of the environment and following the effects of different ATP concentrations in physiological conditions would be relevant, but this is clearly beyond the scope of this study. I recommend publication of the manuscript in its revised form.

Reviewer #2

(Remarks to the Author)

The two main concerns in my original review were that experiments in this study were conducted under nonphysiological ionic strength (10 mM KCl) and that no analysis was performed to test statistical significance of the differences observed under different experimental conditions. The author hasn't attempted to address these concerns in any constructive way, arguing that neither statistical tests nor additional experiments under physiologically relevant buffer conditions are needed. I find this response rather unacceptable. Even though the study is potentially interesting, I cannot recommend publication of this manuscript in the present form.

Specifically:

1. In response to the comment requesting additional experiments at physiologically relevant ionic strength, the author focuses on the concentration of chloride ions, completely ignoring the critical issue of low ionic strength of the buffer used in most of the experiments. It is well established in the literature that condensation of many proteins is highly sensitive to ionic strength of the buffer and even minor variations in salt concentration can lead to markedly different outcomes. Therefore, experimental buffers must be carefully selected, and physiologically relevant salt concentrations and ionic strengths should be used [Alberti S, Gladfelter A, Mittag T. Considerations and Challenges in Studying Liquid-Liquid Phase Separation and Biomolecular Condensates. Cell. 2019]. Without data obtained under physiologically relevant ionic strength, the significance of the current study remains unclear.
2. In response to the comment requesting statistical tests, the author seems to argue that such tests are redundant in biophysical studies, where loosely defined "trends" are more important than rigorous statistical analysis. I strongly disagree with this contention. Statistical tests are necessary to assess the validity of major conclusions of this study.

Version 2:

Reviewer comments:

Reviewer #4

(Remarks to the Author)

In the manuscript, the authors discuss the different effects of ATP to the clustering of FUS, depending on the ATP concentration. In addition, not only ATP but other small molecules are also widely compared, and the results show clear trend that support their findings. Although the mechanism that switches the role of ATP is not totally clear, this is probably beyond the scope of this work. It is true that the ion concentration is lower than the typical physiological condition, but I agree with the reviewer 1 and think that the experiments are reasonable. As for the statistical significance, it seems that even without rigorous analysis the trends in the differences are reasonable. Therefore, I recommend that the manuscript is accepted for publication.

Response to Reviewers' Comments:

Reviewer #1 (Remarks to the Author):

The manuscript describes the effect of varying concentrations of the physiological metabolite ATP on the molecular assembly of FET proteins in vitro. The focus of the work is the formation of mesoscale molecular assemblies (clusters), which form at concentrations well below the saturation threshold for phase separation. These protein family has been associated with neurodegenerative disorders, thereby understanding the effect of metabolites in their aggregation propensity has biomedical relevance. The biophysical study was performed via dynamic light scattering (DLS) and nanoparticle tracking analysis (NTA) using ATP, ADP, AMP and other small amphiphilic molecules to dissect the role of the different chemical groups on the aggregation propensity. FUS-SNAP data were compared with untagged FUS and FUS tagged with other groups to characterize the additional effect of the tag on the aggregation behaviour. The data obtained with all cosolutes and all FUS variants were systematically compared to draw the conclusions (with an appropriate statistical analysis). NanoDSF was also used to characterize the binding of cosolutes to the proteins via changes in thermal protein's stability. The author finds that the phosphate moiety in nucleotides plays a pivotal role in regulating the size of FUS-SNAP clusters and that sequence-specific chemical properties of all cosolute tested play a critical role in determining the effects of small molecules on sub-saturation cluster formation. A complex picture emerges, which highlights the roles of different chemical interactions in the overall clustering response of the FET family. The author suggests three possible scenarios related to the physiological role of ATP, which helps summarizing the large amount of data collected and proposes that sequence-specific interactions between the protein and ATP modulate the cluster formation, challenging conventional hydrotropic and kosmotropic models of action and possibly fostering new studies on the clustering of FET proteins. The manuscript is interesting and timely, and proposes a novel idea on the modulation of the clustering of FET proteins, which is interesting for a general audience.

Response: Thank you very much for your time and for reviewing my manuscript.

Below, two minor points which could be addressed:

1) Besides the average concentrations in different cells, it would be interesting to discuss some known changes of ATP levels in the same cell during cellular homeostasis, which could modulate the formation of the assemblies.

Response to comment 1: Under stable, nutrient-rich conditions, studies in bacteria have shown that ATP levels fluctuate significantly and are partially coordinated with the cell cycle, a complex regulatory pathway (Lin et al., *Current Biology*, 2022). However, such dynamic analyses of ATP levels have not yet been conducted in eukaryotic cells. In this study, I focus on the FET family of proteins, which are known to be prone to aggregation, a process strongly linked to pathological conditions. These proteins primarily reside in the nucleus and are exported to the cytoplasm along with RNA, reflecting their central role in RNA biogenesis. Importantly, their aggregation occurs exclusively in the cytoplasm. Using a bottom-up approach, I aim to understand how ATP influences this process of FET protein aggregation. Although FET proteins

are expressed in nearly all cell types, their pathological aggregation predominantly occurs in neurons. Interestingly, neurons also have the lowest average ATP concentrations in the body, suggesting that ATP availability might play a crucial role in disease onset. Of course, the process might be far more complex; we need to discover by increasing the complexity of our reconstitution experiments. In future studies, I intend to investigate these dynamics under homeostatic conditions in living cells, which will further clarify how cellular energy states influence protein aggregation and neurodegeneration.

2) The effect of pH on the ATP-induced molecular assemblies was not characterized, and the possible role of changes in the local pH in cellular milieu as additional regulator of the aggregation should be mentioned.

Response to comment 2: Thank you very much for this valuable suggestion. I completely agree, and I have now incorporated data for pH levels of 5.5, 7.5, and 8.2 into the revised manuscript.

In my earlier study (Kar et al., PNAS, 2022), we investigated the clustering and phase separation behavior of FUS proteins across different pH values. Those results showed that clustering is observed across all pH conditions, whereas phase separation is highly pH-dependent: at lower pH (~5), phase separation is significantly suppressed, while at higher pH it is greatly enhanced. However, the size of the clusters is pH-dependent, with smaller clusters at lower pH and larger clusters at higher pH—a trend that we have also observed in the present study.

Additionally, we found that the effect of ATP on FUS clustering is significantly reduced at lower pH levels compared to higher pH levels. This is likely due to the pKa (~6.3) of ATP, suggesting that phosphate groups are involved in protein binding in a pH-dependent manner. These new data provide further mechanistic insight (phosphate group interactions with proteins) into the pH dependence of ATP's modulation of FUS clustering, and we have included these findings in the revised manuscript.

Reviewer #2 (Remarks to the Author):

FET (FUS-EWSR1-TAF15) family proteins have the ability to undergo liquid–liquid phase separation (LLPS). Previous studies have shown that FET proteins inherently form mesoscale molecular assemblies, known as clusters, even at concentrations well below the threshold for LLPS. Given the critical role of FET proteins in neurodegenerative diseases, there is growing interest in understanding their LLPS behaviour and cluster formation mechanisms.

This study demonstrates that adenosine triphosphate (ATP)—an amphiphilic molecule and essential cellular metabolite—modulates the size of these sub-saturation mesoscale clusters in a concentration-dependent manner. At lower ATP concentrations, cluster sizes increase, while higher ATP levels lead to reduced cluster size. Other amphiphilic molecules, including common hydrotropes such as sodium xylene sulfonate and sodium toluene sulfonate, exhibit similar concentration-dependent effects on FET protein clustering.

All experiments in this study were conducted under non-physiological conditions and in most cases, proteins had tag. As a result, the biological significance of these findings remains unclear. There are a number of points that need to be addressed by the authors.

Response: Thank you very much for your time and for reviewing my manuscript. Apologies that the reason for the conditions used was not clearly described in the original submission. I have now addressed these concerns and clarified the experimental conditions in the revised manuscript.

Major Concerns:

1. Non-physiological Ionic Strength: All experiments in this study were conducted at low ionic strength (10 mM KCl), which is significantly below physiological levels. The authors state that higher KCl concentrations inhibit FET protein clustering. However, it remains unclear whether physiological salt concentrations merely inhibit clustering or completely abrogate it. If clustering is abrogated under physiological conditions, the relevance of these findings becomes questionable. If only inhibited, to what extent does this occur? What is the effect of ATP and other amphiphilic small molecules on sub-saturation cluster formation under physiological KCl concentrations? This information is critical for assessing the physiological significance of the study.

Response to comment 1: The term “*physiological salt concentration*” has traditionally been used to describe extracellular buffer conditions that match cellular osmolarity (i.e., isotonic conditions). However, inside cells, the concentration of chloride ions is much lower—around 10 mM—according to *Cell Biology by the Numbers* and others. This means that intracellular proteins, such as those from the FET family, are unlikely to encounter the high chloride levels typically used in *in vitro* experiments during their lifetime.

Instead of chloride, cells balance osmotic pressure inside using other anions such as glutamate, phosphate, and macromolecular counterions like proteins. We recently demonstrated that replacing chloride ions with glutamate enhances the clustering

behavior of FET family proteins, without altering their saturation concentration for phase separation (Kar et al., *Nature Commun*, 2023).

In my current bottom-up approach to studying how ATP binds and regulates FET family proteins, I have used KCl at concentrations that reflect the intracellular chloride levels. I have also tested the effects of ATP in the presence of high glutamate concentrations. However, under such conditions, it becomes challenging to disentangle the individual contributions of ATP and glutamate to the clustering of FET proteins. Thus, in this study, I have focused on 20 mM HEPES at pH 7.4 and 10 mM KCl conditions for all experiments.

2. Statistical Significance and Data Interpretation: The manuscript frequently describes observed changes as "slight size increase" (e.g., lines 173–174), without providing statistical analysis to support these claims. Wherever possible, the authors should include statistical significance testing. Specifically, Figures 2 through 8 should include appropriate statistical analyses (e.g., p-values), and the interpretation of results in these figures should be revisited in light of the statistical outcomes.

Response to comment 2: Thank you very much for the suggestion regarding the inclusion of statistical analyses. I agree that statistical measures can be valuable in data interpretation. However, p-values are typically used in contexts such as run charts, normality tests, and regression analyses. In the present work, I have not employed these types of analyses, as the study primarily focuses on biophysical measurements and descriptive trends. While it is certainly possible to analyze the data differently to include p-values, I carefully considered whether this would significantly enhance the understanding of the data presented. Upon reflection, I did not find a clear justification for including p-values in this context, as the trends and conclusions are already well-supported by the current analyses. I hope this clarifies the rationale behind the current data presentation.

3. Impact of SNAP-Tag on Cluster Formation in the presence of HD: The author established that the SNAP-tag can significantly influence FUS cluster formation, particularly in relation to the effects of 1,6-hexanediol (HD). The author demonstrated that the effects of HD are similar for FUS-SNAP and EWSR1-SNAP but differ for TAF15-SNAP. Given the known impact of SNAP-tag on protein behaviour, HD-related experiments should be repeated using untagged proteins to accurately assess the role of HD in cluster formation across the FET family.

4. Use of Tagged Proteins. The protein concentrations used in Figures 2–8 is very low (typically 0.25 μ M), and the sample volumes are small. Given the potential for the SNAP-tag to affect formation of clusters, I would recommend that dynamic light scattering (DLS) or nanoparticle tracking analysis (NTA) experiments—especially those involving ATP, physiologically relevant molecule tested—be conducted using untagged proteins.

Response to comments 3 & 4: Purifying untagged TAF15 and EWSR1 remains highly challenging. For this reason, previous studies, including Patel et al. (*Science*, 2017; *Cell*, 2015), Wang et al. (*Cell*, 2018), and our prior work (Kar et al., *PNAS*, 2022; *Nat. Commun.*, 2024), have utilized tagged proteins for experimental studies. Importantly,

by using a consistent tag across FUS, EWSR1, and TAF15, we ensure that the observed differences reflect the intrinsic, sequence-dependent behaviors of the FET proteins rather than tag-related variability.

We would also like to highlight that, for FUS, we have conducted studies without the SNAP tag, demonstrating its clustering behavior in the presence of small molecules. These experiments did not show significant differences in clustering compared to the SNAP-tagged protein, indicating that the SNAP tag does not substantially alter the clustering properties under our experimental conditions. While we acknowledge that SNAP is a 23 kDa tag and may have some effect on the process, obtaining sufficient quantities of untagged EWSR1 and TAF15 for systematic studies remains technically unfeasible. Thus, the use of SNAP-tagged proteins is necessary to enable comparative studies across the FET family.

Minor Concerns:

1. Lines 141 and 146. The manuscript refers to Figure 2f in both instances, but it should be corrected to Figure 2d.

Response to comment 1: Thank you very much for pointing out the mistakes. I have corrected them.

2. Line 173. The phrase "obscure data" is unclear—please clarify its meaning. Additionally, why are the data for 5 and 10 mM AMP missing in the DLS results (Figures 3c and 3e), while the corresponding NTA data are shown in Figure 3h?

Response to comment 2: At concentrations of 5 and 10 mM AMP, the DLS data were highly noisy, with size distributions ranging from 10 nm to several thousand nm across more than three independent experiments. Moreover, the autocorrelation function values for these measurements were below 0.1, indicating a very low signal-to-noise ratio. Consistently, NTA data at 10 mM AMP showed that the cluster populations were extremely low. For these reasons, the DLS data at these concentrations were excluded from the plot, as they were too unreliable for meaningful interpretation. Which I explained as 'obscure'.

3. In Figure 7, the author uses 0.3 μM EWSR1-SNAP, whereas 0.25 μM protein concentrations are used elsewhere (e.g., for FUS-SNAP, FUS, and TAF15-SNAP). The rationale for using a different concentration for EWSR1-SNAP should be clarified, as this inconsistency complicates direct comparison across proteins.

Response to comment 3: Apologies for not explaining this anomaly. I have added a note in the DLS measurements stating the following. "The proteins were purified and stored in high-salt (500 mM KCl) buffer, which contributes background salt to the final KCl concentration in the experiments. For FUS-SNAP, the stock solution can be concentrated to $\sim 50 \mu\text{M}$, providing a broader range of concentrations for experiments. However, for untagged FUS, EWSR1-SNAP, and TAF15-SNAP, the stock concentrations remain below 20 μM . To maintain a final KCl concentration of $\sim 10 \text{ mM}$ under sub-saturation conditions, we use lower protein concentrations in experiments, such as 0.25 μM for untagged FUS and TAF15-SNAP, and 0.3 μM for EWSR1-SNAP."

4. The volume fraction of clusters for both FUS-SNAP and EWSR1-SNAP increases at 1 mM ATP and decreases at 5 mM ATP, indicating a similar response. However, EWSR1-SNAP contains 10 more arginine residues than FUS-SNAP. Conversely, TAF15-SNAP also contains 10 more arginines than EWSR1-SNAP but requires significantly higher ATP concentrations (15–20 mM) to observe a similar decrease in cluster size. How does author reconcile the similar ATP responses between FUS-SNAP and EWSR1-SNAP despite their arginine content differences, while TAF15-SNAP, with a comparable arginine difference relative to EWSR1-SNAP, shows a markedly different behaviour?

Response to comment 4: I apologize for drawing these conclusions (regarding arginine-phosphate interactions) more strongly than I intended. I agree with your assessment that the correlation is not strong. The full-length proteins are capable of various non-specific interactions. My intention was to highlight that, among the multiple potential non-specific interactions, arginine-phosphate binding is one possible contributor that correlates with the observed data.

I have revised the paragraph accordingly in the manuscript, also incorporating additional suggestions from the other reviewer to improve clarity and balance in the discussion.

Reviewer #3 (Remarks to the Author):

The author of “Molecular Matchmakers: How ATP and Small Amphiphilic Molecules Fine-Tune FET Protein Clusters” explores how nucleotides with varying amounts of phosphate as well as hydrotropes and the amphiphilic molecule 1,6-hexanediol influence the propensity of sub-micron protein clusters of FET proteins to form. With the use of dynamic light scattering and nanoparticle tracking analysis the size of clusters is monitored over time for each solute, showing that clusters generally grow in size and saturate around 30 minutes. Cluster formation depends on the solute concentration, where increased solute decreases cluster size or inhibits cluster formation completely through disruption of pi-pi and electrostatic interactions. The author concludes that this behavior comes from sequence specific interactions.

There is potential for this work to be of broad interest, but the interpretation of the data as well as the conclusions are the same as the previously published paper Patel, Science, 2017, that investigated the hydrotrope effect of many of the same molecules as in this work. The findings here do not provide new insight into the molecular mechanism compared to the previous literature. The main new point of the work is that the hydrotrope effect is observed at the cluster sub micron size, compared to the mesoscopic super micron condensate, which is not surprising. It is recommended that major revisions are made to the manuscript

Response: Thank you very much for taking the time to review the manuscript. I would like to clarify that the present work is distinct from Patel et al.'s Science paper for the following reasons:

1. Different conclusions regarding hydrotropy: While Patel et al. first described ATP as a ‘biological hydrotrope’, here I report that hydrotropic behavior alone does not explain the observed mesoscale cluster formation.

2. The Science paper focused solely on the effects of ATP on phase-separated condensates. In contrast, this work investigates the effect of ATP on mesoscale clusters, which are distinctly different entities from phase-separated condensate, as demonstrated in our previous work (Kar et al., PNAS, 2022; Kar et al., Nat Commun, 2024).

3. Patel et al. describe that increasing concentrations of ATP solubilize the phase-separated condensates using microscopic estimations, which have resolution disadvantages. In my earlier work (Kar et al., PNAS 2022), I reported that ATP indeed dissolves the phase-separated condensates; however, it does not solubilize the mesoscale clusters. In this present work, I have systematically used all other small amphiphilic molecules, including ATP, to understand how the mesoscale clusters are modulated by these small molecules.

Major concerns:

- 1) Three techniques are used throughout the paper, two of which track particle size. These methods provide no insight into molecular mechanisms of ATP-protein interactions. The manuscript would be strengthened through the addition of complementary spectroscopic measurements, such as NMR or EPR, or vibrational

techniques such as two dimensional infrared, that provide informational on structure and dynamics of protein complexes. This would especially shed light onto what sequence specific interactions are important for the regulation of cluster size

Response to comment 1: Thank you very much for this valuable suggestion. Indeed, it was on my list to do, as these approaches would provide important insights. However, this approach also brings challenges to studying mesoscale clusters. Both EPR and NMR studies require substantial amounts of material, which is challenging within the scope of this study.

Since the discovery of mesoscale clusters (Kar et al., PNAS, 2022), we have explored the use of NMR and EPR to probe molecular interactions. However, obtaining interpretable data using these techniques has proven unfeasible because mesoscale clusters form under highly dilute conditions, while NMR and EPR require high sample concentrations. Increasing the concentration to meet these requirements shifts the system into the phase-separated regime, rather than the clustering regime examined in this study.

While phase-separated condensates have been extensively studied using NMR, the current work specifically focuses on the distinct properties of mesoscale clusters, which are fundamentally different from phase-separated condensates. Nevertheless, I believe that similar interaction landscapes (as shown in Kar et al, PNAS 2022) may underlie both systems, and we have cited relevant NMR study references to support and contextualize our data accordingly.

2) Data provided in the manuscript is comprehensive, but to the point that it is too dense. Figures are small and often hard to read, especially those with insets, and panels within a figure often have repetitive data. Shifting some data to the supporting information and increasing figure size would improve accessibility of the data.

Response to comment 2: I have separated the insets data from the NTA plots and transferred it to the SI.

3) For some data sets, concentrations are missing or are reported as inconclusive and left out.

Response to comment 3: I believe you are referring to the data on AMP effects on FUS-SNAP clustering, where the results for 5 and 10 mM AMP are not presented in the main figures. At concentrations of 5 and 10 mM AMP, the DLS data were highly noisy, with size distributions ranging from 10 nm to several thousand nm across more than three independent experiments. Moreover, the autocorrelation function values for these measurements were below 0.1, indicating a very low signal-to-noise ratio. Consistently, NTA data at 10 mM AMP showed that the cluster populations were extremely low. For these reasons, the DLS data at these concentrations were excluded from the plot, as they were too unreliable for meaningful interpretation. Which I explained as 'obscure'.

4) It is concluded in the manuscript that ATP is not a kosmotrope due to its concentration dependent behavior. However, it seems the behavior observed here is consistent with kosmotropes as they often exhibit nonlinear behavior with

concentration where the protein is stabilized at low concentration and is destabilized at high concentrations. Additionally, many kosmotropes/chaotropes have non-covalent interactions with other molecules through solvent shared ion pairs or contact-ion pairs, and their behavior is unique to the specific surface chemistry of the protein.

Response to comment 4: According to the literature, the Hofmeister effect (kosmotropic and chaotropic effects) of different ions on aqueous solution properties, by definition, concerns only ion solvation effects (Jungwirth et al., Nat. Chem., 2014). However, it has long been discussed that a key limitation of this framework is its inability to account for specific surface chemistry and molecular interactions with proteins. In the present work, my aim is to move beyond the Hofmeister and hydrotropic perspectives by specifically examining how small molecules interact directly with proteins, thereby providing a deeper understanding of their modulatory roles that cannot be fully explained by general ion solvation theories.

Minor Concerns

1) The number of times data were reproduced is not mentioned in the manuscript

Response to comment 1: I would like to draw the reviewer's attention to the figure captions, where the final sentences explicitly state the number of individual repeats conducted for each dataset presented.

2) Autocorrelation functions of DLS with no slow modes are referred to throughout the text, but how this conclusion is made from the plotted data is unclear.

Response to comment 2: Apologies for making unclear statements. It was my mistake not to include the proper reference for the statements. In my earlier work (Kar et al., PNAS 2022), we have shown that the cluster coalescence (arrow indicating slow modes) can be studied in real-time through the autocorrelation function of DLS. At phase separation concentrations and above, the slow modes appear, suggesting cluster coalescence. In contrast, the clusters at lower concentrations are stable in the same time frame. I have added the reference to the text.

3) Error of the melting temperature is not reported

Response to comment 3: The error bars are included in the plots, represented as grey shaded areas around the lines.

4) Discussion of redshift and blueshift of NanoDSF data in terms of solvent exposed/solvent buried would make the conclusions more accessible to a broader audience

Response to comment 4: I have added clarifications regarding “solvent-exposed” and “solvent-buried” residues in the text to improve clarity.

Reviewers' comments:

Reviewer #1 (Remarks to the Author):

The author has addressed my concerns in the revised version of the manuscript. The additional data on the pH dependency of the ATP-induced molecular assemblies provide further insights into the process of molecular assembly of FET proteins in vitro, and shed light on the role of the phosphate groups of ATP in the process. The author agrees that increasing the complexity of the environment and following the effects of different ATP concentrations in physiological conditions would be relevant, but this is clearly beyond the scope of this study. I recommend publication of the manuscript in its revised form.

Thank you very much for your recommendations on publishing the revised manuscript.

Reviewer #2 (Remarks to the Author):

Concern: The two main concerns in my original review were that experiments in this study were conducted under nonphysiological ionic strength (10 mM KCl) and that no analysis was performed to test statistical significance of the differences observed under different experimental conditions. The author hasn't attempted to address these concerns in any constructive way, arguing that neither statistical tests nor additional experiments under physiologically relevant buffer conditions are needed. I find this response rather unacceptable. Even though the study is potentially interesting, I cannot recommend publication of this manuscript in the present form.

Specifically:

1. In response to the comment requesting additional experiments at physiologically relevant ionic strength, the author focuses on the concentration of chloride ions, completely ignoring the critical issue of low ionic strength of the buffer used in most of the experiments. It is well established in the literature that condensation of many proteins is highly sensitive to ionic strength of the buffer and even minor variations in salt concentration can lead to markedly different outcomes. Therefore, experimental buffers must be carefully selected, and physiologically relevant salt concentrations and ionic strengths should be used [Alberti S, Gladfelter A, Mittag T. Considerations and Challenges in Studying Liquid-Liquid Phase Separation and Biomolecular Condensates. *Cell*. 2019]. Without data obtained under physiologically relevant ionic strength, the significance of the current study remains unclear.

Comment: I agree that ionic strength is important in biochemical studies. However, most biochemical work has traditionally relied on "physiological buffer" mimics that closely resemble extracellular-like buffers that maintain osmotic pressure but do not accurately reflect intracellular ionic conditions. It has long been known that cells contain lower chloride ions (*Cell Biology by the Numbers*). At higher concentrations, chloride binds to the protein backbone and suppresses protein-protein interactions (Sengupta et al., *Biochemistry* 2016; Kar et al., *Nat. Commun.* 2024).

By contrast, potassium glutamate (Kglu) more closely mimics intracellular conditions and does not bind to protein backbones, making it a better proxy for physiological buffers (Cheng et al., *Biophys. J.* 2016). Indeed, Kglu strongly enhances clustering of FET proteins compared to

equivalent KCl concentrations (Kar et al., Nat. Commun. 2024). Cellular Kglu levels also vary between 5–70 mM, depending on cell type (Kar et al., Nat. Commun. 2024).

In this study, my aim was to dissect how ATP and amphiphilic small molecules interact with FET proteins. To achieve this, I used ~10 mM KCl, which reflects intracellular chloride levels and yields FET clustering propensities similar to those observed at ~100 mM Kglu. This choice avoids confounding effects from Kglu itself and allows a clearer interpretation of small molecule–protein interactions.

In summary, I intentionally compromised slightly on “ionic strength” to better approximate intracellular ionic composition and thereby obtain mechanistic insights into how FET proteins respond to small molecules.

2. In response to the comment requesting statistical tests, the author seems to argue that such tests are redundant in biophysical studies, where loosely defined “trends” are more important than rigorous statistical analysis. I strongly disagree with this contention. Statistical tests are necessary to assess the validity of major conclusions of this study.

Comment 2: This study comprises seven different small molecules tested at six concentrations each. Conventional statistical significance plotting is not feasible for the data presented in the current manuscript. I am willing to implement alternative, statistically sound approaches if concrete suggestions are provided, but a standard significance plot would be inappropriate for the present graphs.

Round 3

Response to Reviewers' Comments:

Reviewer #4 (Remarks to the Author):

Concern: In the manuscript, the authors discuss the different effects of ATP to the clustering of FUS, depending on the ATP concentration. In addition, not only ATP but other small molecules are also widely compared, and the results show clear trend that support their findings. Although the mechanism that switches the role of ATP is not totally clear, this is probably beyond the scope of this work. It is true that the ion concentration is lower than the typical physiological condition, but I agree with the reviewer 1 and think that the experiments are reasonable. As for the statistical significance, it seems that even without rigorous analysis the trends in the differences are reasonable. Therefore, I recommend that the manuscript is accepted for publication.

Comment: I sincerely thank the reviewer for the positive and thoughtful evaluation of this work. I appreciate their recognition of the clear trends in ATP and small-molecule–dependent clustering of FUS, as well as their understanding that elucidating the complete mechanistic switching behavior of ATP is beyond the present scope. I also thank the reviewer for acknowledging the reasonableness of our experimental conditions and conclusions.